# Credal Prediction based on Relative Likelihood

**Timo Löhr**[*]
LMU Munich, MCML
`timo.loehr@ifi.lmu.de`

**Paul Hofman**[*]
LMU Munich, MCML
`paul.hofman@ifi.lmu.de`

**Felix Mohr**
Universidad de La Sabana
`felix.mohr@unisabana.edu.co`

**Eyke Hüllermeier**
LMU Munich, MCML, DFKI
`eyke@lmu.de`

## Abstract

Predictions in the form of sets of probability distributions, so-called credal sets, provide a suitable means to represent a learner's epistemic uncertainty. In this paper, we propose a theoretically grounded approach to credal prediction based on the statistical notion of relative likelihood: The target of prediction is the set of all (conditional) probability distributions produced by the collection of plausible models, namely those models whose relative likelihood exceeds a specified threshold. This threshold has an intuitive interpretation and allows for controlling the trade-off between correctness and precision of credal predictions. We tackle the problem of approximating credal sets defined in this way by means of suitably modified ensemble learning techniques. To validate our approach, we illustrate its effectiveness by experiments on benchmark datasets demonstrating superior uncertainty representation without compromising predictive performance. We also compare our method against several state-of-the-art baselines in credal prediction.

## 1 Introduction

The distinction between two types of uncertainty, referred to as aleatoric and epistemic, is receiving increasing interest in machine learning [Hüllermeier and Waegeman, 2021]. Roughly speaking, the aleatoric uncertainty of a predictive model is caused by the inherent randomness of the data-generating process, whereas epistemic uncertainty is caused by the learner's lack of knowledge about the true (or best) predictive model. While aleatoric uncertainty is irreducible, epistemic uncertainty can be reduced on the basis of additional information, e.g., by collecting more training data.

Aleatoric uncertainty can be captured adequately in terms of (conditional) probability distributions, whereas the representation of epistemic uncertainty requires "second-order" formalisms more general than (single) probability distributions. In the Bayesian approach to machine learning, second-order probability distributions are used for this purpose [Depeweg et al., 2018, Kendall and Gal, 2017]: The learner maintains a probability distribution over the model space, which, if each model is a probabilistic predictor, results in predictions in the form of probability distributions of probability distributions (on outcomes). The Bayesian approach is theoretically appealing but computationally demanding, and commonly criticized for the need to specify a prior distribution (strongly influencing prediction and uncertainty quantification) [Gawlikowski et al., 2023].

An alternative second-order formalism is offered by *credal sets* [Walley, 1991], i.e., sets (instead of distributions) of probability distributions that are commonly assumed to be closed and convex [Cozman, 2000]. Although a set appears to provide weaker information than a distribution, a set-based representation also has advantages. In particular, it can be argued that sets are more apt at

---

[*]equal contribution

39th Conference on Neural Information Processing Systems (NeurIPS 2025).

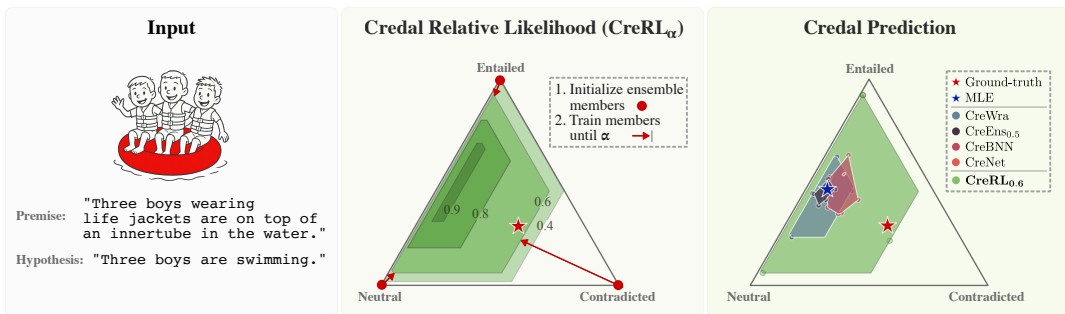

**Figure 1: Overview of our method**. Language inference example from the ChaosNLI dataset, showing a premise-hypothesis pair with three possible labels: *Entailed*, *Neutral* and *Contradicted*. Middle: Illustration of the learning mechanism of our Credal Relative Likelihood (CreRL) framework, showing ensemble member training up to different $\alpha$ thresholds. Right: Unlike baselines, which are centered around the MLE and remain fixed, our method allows to adapt credal set size to improve coverage of the ground-truth distribution.

representing *ignorance* in the sense of a lack of knowledge [Dubois et al., 1996], because distributions always involve additional assumptions beyond the mere distinction between plausible and implausible candidate models. Decision-making on the basis of credal sets has been proposed as a reasonable alternative to Bayesian decision-making [Levi, 1978, Girón and Ríos, 1980]. The learning of credal predictors (i.e., models making predictions in the form of credal sets) has been considered in machine learning in the past [Zaffalon, 2001, Corani and Zaffalon, 2008] and, in light of the quest for proper representation of epistemic uncertainty, received renewed interest more recently [Wang et al., 2024a, Nguyen et al., 2025, Cella and Martin, 2024].

A major question to be addressed in this regard concerns the distinction between plausible and implausible models. In this paper, we adopt a theoretically grounded approach for constructing credal sets based on the notion of relative likelihood. Our approach explicitly defines the prediction target as the set of all probability distributions generated by plausible learners, namely those whose relative likelihood exceeds a specified threshold (cf. Section 3). That said, realizing this in complex machine learning settings comes with several practical challenges, notably the questions of representation, approximation, and inference: How to formally represent subsets of a high-dimensional model space, how to approximate such sets algorithmically, and how to infer credal predictions from them? We propose methods to address these questions based on suitably modified ensemble techniques (cf. Section 4). We highlight the effectiveness of our approach on real-world datasets, evaluating its performance in terms of coverage and efficiency as key criteria of credal prediction. Also, we analyze its performance on downstream tasks such as Out-of-Distribution (OoD) detection (cf. Section 5).

**Contributions.** In summary our contributions are as follows:

1. We develop a theoretically sound approach to learning credal predictors, which is grounded in the statistical notion of relative likelihood.

2. We cast the learning task as optimizing multi-objective generalization performance, namely as finding a compromise between coverage and efficiency of credal predictions.

3. We propose an adaptive and conceptually intuitive ensemble-based method for approximating sets of plausible models and the induced credal predictions.

4. We empirically compare our method to representative baselines and, for the first time, compare state-of-the-art credal predictors by coverage and efficiency. Our approach achieves superior uncertainty representation and strong OoD performance.

**Related Work.** In machine learning, uncertainty is often represented by (approximate) Bayesian methods [Mackay, 1992, Blundell et al., 2015, Lakshminarayanan et al., 2017, Gal and Ghahramani, 2016, Daxberger et al., 2021]. An important characteristic of such representations, especially for uncertainty tasks, is diversity [D'Angelo and Fortuin, 2021, Wood et al., 2023], which can be enforced by regularization [de Mathelin et al., 2023], varying hyper-parameters [Wenzel et al., 2020], or increasing diversity in representations [Lopes et al., 2022]. Alternatively, credal sets have been

used in the fields of imprecise probability and machine learning to represent model uncertainty [Zaffalon, 2001, Corani and Zaffalon, 2008, Corani and Mignatti, 2015]. Such sets can be generated based on the relative likelihood, also referred to as normalized likelihood [Antonucci et al., 2012a] and have been used in machine learning with simple model classes [Senge et al., 2014, Cella and Martin, 2024]. In this work, we build upon these approaches and address the challenges that emerge when adapting the relative likelihood to a setting with complex predictors. Recently, credal sets have also been applied in the context of deep learning. Some approaches use ensemble learning to derive class-wise lower and upper probabilities [Wang et al., 2024b, Nguyen et al., 2025]. Others train models to directly predict probability intervals [Wang et al., 2024a]. Hybrid methods, combining multiple uncertainty frameworks, have also been proposed. Caprio et al. [2023] combine Bayesian deep learning and credal sets by considering sets of priors. Another approach leverages conformal prediction to construct credal sets with validity guarantees [Javanmardi et al., 2024]. A more detailed discussion of the related work is provided in Appendix A.

## 2 Problem Statement

We consider classification in a supervised learning setting. We have access to training data $\mathcal{D} = \{(\boldsymbol{x}_i, y_i)\}_{i=1}^N \subset \mathcal{X} \times \mathcal{Y}$, where $\mathcal{X}$ is the instance space and $\mathcal{Y} = \{y_1, \ldots, y_K\}$ is the label space with $K \in \mathbb{N}$ classes. The training data are realizations of random variables that are independently and identically distributed according to a probability measure on $\mathcal{X} \times \mathcal{Y}$. We consider a hypothesis space $\mathcal{H}$ with probabilistic predictors $h : \mathcal{X} \to \Delta_K$, where $\Delta_K$ denotes the $(K-1)$-simplex (that we will also refer to as the probability simplex). Given an instance $\boldsymbol{x}$, a predictor $h$ assigns a probability distribution $p(\cdot \mid \boldsymbol{x}, h) = h(\boldsymbol{x})$, which is an estimate of the ground-truth probability $p(\cdot \mid \boldsymbol{x}, h^*)$ generated by the ground-truth model $h^*$.

Predictive models $h$ are often evaluated in terms of their likelihood $L(h) = \prod_{i=1}^N p(y_i \mid \boldsymbol{x}_i, h)$. Then, adopting the established principle of maximum likelihood inference, the model of choice is the maximum likelihood estimator (MLE) $h^{ML}$, i.e., the model whose likelihood is highest. By predicting probability distributions $h^{ML}(\boldsymbol{x})$ on $\mathcal{Y}$, this model is able to represent *aleatoric* uncertainty. However, it cannot capture information about the uncertainty of $h^{ML}$ itself, i.e., about how much it possibly deviates from the ground-truth model $h^*$.

In order to capture this *epistemic* uncertainty, we consider a second-order uncertainty representation in the form of credal sets: Instead of relying on a single predictor $h$, the uncertainty about the true underlying model $h^*$ is represented by the set of plausible predictors $\mathcal{C} \subseteq \mathcal{H}$. For a given query instance $\boldsymbol{x} \in \mathcal{X}$, this set induces a prediction in the form of a credal set of distributions:

$$\mathcal{Q}_{\boldsymbol{x}} = \{p(\cdot \mid \boldsymbol{x}, h) : h \in \mathcal{C}\} \subseteq \Delta_K. \tag{1}$$

We define the problem of learning a credal predictor as a generalization of the standard setting of supervised learning as introduced above: Given training data $\mathcal{D}$, the task is to induce a model $H : \mathcal{X} \to 2^{\Delta_K}$ that delivers predictions in the form of credal sets $\mathcal{Q}_{\boldsymbol{x}} = H(\boldsymbol{x}) \subseteq \Delta_K$. In our setup, $H(\boldsymbol{x})$ represents the mapping of an instance $\boldsymbol{x} \in \mathcal{X}$ through a set of plausible models $\mathcal{C}$ to the corresponding credal set $\mathcal{Q}_{\boldsymbol{x}}$. In general, $H$ could also be realized differently. Inspired by other set-valued prediction methods such as conformal prediction [Vovk et al., 2005], we evaluate such a predictor in terms of its coverage and efficiency. Coverage (of the ground-truth distribution by the cedal set) is defined as follows:

$$C(H) = \mathbb{E}\left[\llbracket p(\cdot \mid \boldsymbol{x}, h^*) \in H(\boldsymbol{x}) \rrbracket\right], \tag{2}$$

where $\llbracket \cdot \rrbracket$ denotes the indicator function and the expectation is taken with regard to the marginal distribution of $\boldsymbol{x}$ on $\mathcal{X}$. Moreover, efficiency captures the idea that "small" (more informative) credal sets are preferred over "large" (less informative) ones. It can be measured in different ways, for example as follows:

$$E(H) = 1 - \mathbb{E}\left[\frac{1}{K}\sum_{k=1}^K \overline{p}(y_k \mid \boldsymbol{x}) - \underline{p}(y_k \mid \boldsymbol{x})\right], \tag{3}$$

where $\overline{p}(y_k \mid \boldsymbol{x}) = \sup\{p(y_k \mid \boldsymbol{x}) : p \in H(\boldsymbol{x})\}$ (and $\underline{p}(y_k \mid \boldsymbol{x})$ is defined analogously). We assume efficiency to be positively oriented, meaning that higher efficiency corresponds to smaller credal sets. As opposed to the volume of a credal set, which, besides being challenging to compute, is not intuitive in high dimensions, this measure of efficiency is particularly interpretable. Specifically, it describes

the (complement of the) average interval length for each class. Importantly, we use efficiency strictly as an indicator of set size, not as a measure of epistemic uncertainty (cf. Section 2).

The empirical coverage and efficiency is determined in terms of the corresponding averages on a finite set of (test) data $\boldsymbol{x}_1, \ldots, \boldsymbol{x}_N$. Note that ground-truth (first-order) distributions $p(\cdot \mid \boldsymbol{x}_i, h^*)$ are typically unavailable during both training and testing, which often necessitates approximating coverage through alternative means. In this work, however, we make use of data that provides access to ground-truth distributions.

Evaluating a credal predictor in terms of its coverage and efficiency means that two predictors $H$ and $H'$ are not necessarily comparable in the sense that one of them is "better" than the other one. Instead, such predictors are only comparable in a Pareto sense: $H$ is better than $H'$ if $C(H) \geq C(H')$ and $E(H) \geq E(H')$ (and one of the inequalities is strict). The task can then be specified as learning Pareto-optimal credal predictors.

## 3 Relative Likelihood-Based Credal Sets

To learn a credal predictor $H : \mathcal{X} \to 2^{\Delta_K}$ from training data, we adopt the representation given in (1), defining $\mathcal{C}$ as a set of plausible (first-order) predictors $h : \mathcal{X} \to \Delta_K$ drawn from an underlying hypothesis space $\mathcal{H}$. To characterize plausibility, we employ the concept of relative likelihood:

$$\gamma(h) = \frac{L(h)}{L(h^{ML})} = \frac{L(h)}{\sup\limits_{h' \in \mathcal{H}} L(h')}.$$

Here, $h^{ML}$ represents the model in $\mathcal{H}$ with the highest likelihood given the training data $\mathcal{D}$. This notion, also called extended or normalized likelihood, has been proposed by Birnbaum [1962] and used for statistical inference [Wasserman, 1990, Walley and Moral, 1999]. It offers an attractive alternative to Bayesian or frequentist reasoning eliminating the need to specify priors or perform resampling [Giang and Shenoy, 2002]. In machine learning this is especially useful as specifying a meaningful prior on $\mathcal{H}$ is a non-trivial problem and resampling from the data distribution is usually not possible either, as we only have access to the given samples $\mathcal{D}$. An axiomatic justification of the relative likelihood (in the context of evidence theory) has been given by Denoeux [2014], who derives it from (i) the likelihood principle, (ii) compatibility with the Bayes rule, and (iii) the minimal commitment principle.

On the basis of the relative likelihood, a set of models can be constructed, analogous to a confidence region [Aitkin, 1982], by including models that are plausible in the sense of surpassing a threshold $\alpha \in [0, 1]$. This set of models is also referred to as an $\alpha$-cut [Antonucci et al., 2012a]:

$$\mathcal{C}_\alpha = \{ h : \gamma(h) \geq \alpha \} \subseteq \mathcal{H}. \qquad (4)$$

According to this definition, a model $h$ is considered implausible (and hence ignored) if its likelihood is too small compared to the likelihood of the (presumably) best model, namely if its likelihood is less than $\alpha$ times the likelihood of the best model. Thus, the parameter $\alpha$ has a quite intuitive meaning. The concept of $\alpha$-cuts is illustrated in Figure 2.

Given an instance $\boldsymbol{x} \in \mathcal{X}$, the set of predictors maps to a set of (conditional) distributions:

$$\mathcal{Q}_{\boldsymbol{x}, \alpha} = \{ p(\cdot \mid \boldsymbol{x}, h) : h \in \mathcal{C}_\alpha \} \subseteq \Delta_K.$$

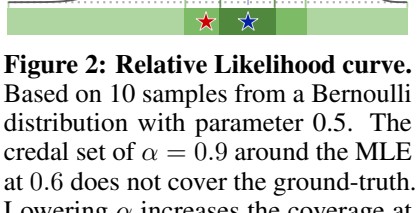

**Figure 2: Relative Likelihood curve.** Based on 10 samples from a Bernoulli distribution with parameter 0.5. The credal set of $\alpha = 0.9$ around the MLE at 0.6 does not cover the ground-truth. Lowering $\alpha$ increases the coverage at the cost of efficiency.

This credal set can then be used for predictive tasks and to quantify uncertainty. In essence, it can be used to express the amount of uncertainty one has about the best predictor that can be obtained from the given data sample.

Considering that we aim to simultaneously optimize coverage and efficiency, it is important to highlight the role of the parameter $\alpha$ as a means to move along the Pareto front and trade coverage against efficiency: The lower $\alpha$, the higher coverage tends to be. At the same time, however, efficiency will deteriorate (cf. again Figure 2). As $\alpha$ increases, the intervals for given data points shrink since

only models with high relative likelihood for the given data sample are included. Conversely, lowering $\alpha$ allows for the inclusion of models with lower likelihood, resulting in larger credal intervals. This controllability induced by $\alpha$ emphasizes the appeal of the relative likelihood as a tool for (plausible) model selection and, thereby, credal set construction.

## 4 Approximating Relative Likelihood-Based Credal Sets

Building on the theoretical framework for constructing credal sets using relative likelihood, it becomes evident that, in practice, we must approximate the set of models $\mathcal{C}_\alpha$, and consequently the resulting credal set $\mathcal{Q}_{\boldsymbol{x},\alpha}$. In this section, we examine how the formal notion of relative likelihood can be applied to construct credal sets in high-dimensional machine learning settings. Up to this point, we made no assumptions about the specific model class. Obviously, to approximate the $\alpha$-cut (4), the model class must allow to define and optimize toward a target relative likelihood during training. As a concrete and widely used example of complex models in machine learning, we focus on neural networks throughout the remainder of this work.

**Estimating the Maximum Likelihood Estimator.** Several strategies can be employed to obtain $h^{ML}$, including training a model with parameters known to perform well, utilizing a pre-trained model, or leveraging AutoML techniques to determine optimal parameter settings during training. Throughout this work, we adopt the first approach and train the maximum likelihood estimator $h^{ML}$ using parameter configurations that have been identified in the literature as effective. Further details and configurations of experiments are provided in Appendix B.

Given an estimated maximum likelihood model, the estimated relative likelihood is obtained as $\hat{\gamma}(h) = \frac{L(h)}{L(h^{ML})}$.

---

**Algorithm 1** Train Credal Relative Likelihood ensemble.

---

**Require:** $\alpha, M$
1: **Step 1: Approximate maximum likelihood model**
2: $h^{ML} \leftarrow \arg\min_h \mathcal{L}(h)$
3: **Step 2: Train ensemble members**
4: $\Delta\tau = \frac{1-\alpha}{M}$
5: **for** $i \in \{0, \ldots, M-1\}$ **do**
6: $\quad \tau_i = \alpha + i \cdot \Delta\tau$
7: $\quad h_i \leftarrow$ ToBias initialization
8: $\quad$ Train $h_i$ such that $\hat{\gamma}(h_i) \approx \tau_i$
9: **end for**

---

**Approximating the $\alpha$-cut.** Having found a suitable candidate model to compute the maximum likelihood, the next step is to construct the $\alpha$-cut. A straightforward approach is to train an ensemble of hypotheses and construct the $\alpha$-cut based on these hypotheses. In essence, approaches based on regular ensemble training [Wang et al., 2024b, Nguyen et al., 2025] form an example of this strategy. However, an issue is that all hypotheses $h$ typically tend to cluster around $h^{ML}$, hence the obtained hypotheses do not accurately approximate the $\alpha$-cut (unless $\alpha \approx 1$, see Figure 1).

To obtain better coverage of $\mathcal{C}_\alpha$, we propose to train $M$ hypotheses using an early stopping strategy, namely until a specific relative likelihood value $\tau_i$ is reached. Given $\alpha$, we consider thresholds $\tau = \{\tau_i \mid \tau_i = \alpha + i \cdot \Delta\tau\}$, where $\Delta\tau = \frac{1-\alpha}{M-1}$ and $i \in \{0, \ldots, M-2\}$. This guarantees a broad range of hypotheses in terms of relative likelihoods. The influence of the ensemble size $M$ is analyzed in an ablation study in Appendix C.2.

Although this should cover the $\alpha$-cut well, there might still be a lack of diversity in the predictions of the resulting hypotheses. In the literature, it is known that training ensembles without explicitly encouraging diversity actually minimizes diversity [Abe et al., 2022]. To encourage diversity, we introduce a novel initialization strategy called ToBias. The idea is to make sure that the initial state of the ensemble, i.e., prior to training, represents a state of full uncertainty (or no knowledge). Hence, the resulting credal prediction should entail the entire probability simplex. As the predictors in the ensemble are trained, the amount of knowledge increases, and the predicted credal set should shrink. To enforce this in a finite predictor scenario, we ensure that the initial predictions of the learners correspond to degenerate probability distributions at vertices of the $(K-1)$-simplex, effectively covering the entire simplex in the initially predicted credal set. Figure 1 illustrates the mechanism of ToBias initialization and the overall learning process.

Technically, the initialization is performed by assigning a large constant to one of the biases in the final layer of each predictor. For example, given the biases $\boldsymbol{b}_i = [b_{i,1}, ..., b_{i,K}]$ of the last layer of predictor $h_i$, we set $b_{i,\ i \bmod K} = \beta$, where $\beta$ is large constant, following the regular initialization. In the remainder of this work, we use $\beta = 100$. An ablation study examining the influence of ToBias initialization is presented in Appendix C.2.

We illustrate the difference between regular initialization and ToBias initialization in Figure 3. Observe that without ToBias initialization, the credal set initially represents a state of low epistemic uncertainty, as all predictions are concentrated around the barycenter of the probability simplex, and increases its uncertainty as training progresses. The opposite can be observed with ToBias initialization. The credal set starts representing a state of full uncertainty and decreases its uncertainty as the learners acquire more knowledge.

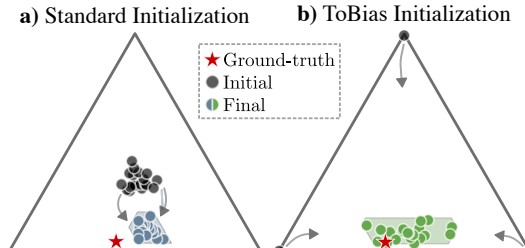

**Figure 3: Effect of ToBias**. Example from ChaosNLI. a) Standard initialization initially predicts close to uniform distributions and converges to a small credal set. b) ToBias initializes at the vertices resulting in diverse predictions.

Having trained $M$ models until the respective thresholds, the approximate $\alpha$-cut $\tilde{\mathcal{C}}_\alpha$ can be constructed as follows:

$$\tilde{\mathcal{C}}_\alpha = \{h_i \ : \hat{\gamma}(h_i) \approx \tau_i\}_{i=0}^{M-1} \subseteq \mathcal{H},$$

where $\hat{\gamma}(h_i) \approx \tau_i$ is enforced by the training process described above. The full algorithm to train Credal Relative Likelihood (CreRL) ensembles is presented in Algorithm 1.

**Credal Set Predictions.** Given a query instance $\boldsymbol{x} \in \mathcal{X}$, the set $\tilde{\mathcal{C}}_\alpha$ induces a finite approximation of the credal set:

$$\tilde{\mathcal{Q}}_{\boldsymbol{x},\alpha} = \{p(\cdot \,|\, \boldsymbol{x}, h) : h \in \tilde{\mathcal{C}}_\alpha\} \subseteq \Delta_K.$$

There are several principled ways of transforming this finite set into a credal set. Two common approaches are to derive (i) the convex hull of the set $\tilde{\mathcal{Q}}_{\boldsymbol{x},\alpha}$ and (ii) the intervals (lower and upper probabilities) for individual classes obtained from this set. Note that the convex hull is contained in the "box" credal set induced by the intervals. In the remainder of this work, we opt for the latter approach and define the approximation of $\mathcal{Q}_{\boldsymbol{x},\alpha}$ as follows [de Campos et al., 1994]:

$$\hat{\mathcal{Q}}_{\boldsymbol{x},\alpha} = \{p(\cdot \,|\, \boldsymbol{x}) : p(y_k \,|\, \boldsymbol{x}) \in [\underline{p}(y_k \,|\, \boldsymbol{x}), \overline{p}(y_k \,|\, \boldsymbol{x})], \forall y_k \in \mathcal{Y}\}$$

## 5 Empirical Results

To empirically evaluate our method, we measure its performance in terms of coverage and efficiency, as defined in Section 2, along with the corresponding Pareto criterion. Additionally, we evaluate our approach on the downstream task of OoD detection, which is often used to assess the quality of (epistemic) uncertainty representations. We compare our approach (CreRL) to suitable baselines by implementing the following approaches: Credal Wrapper (CreWra) [Wang et al., 2024b], Credal Ensembling (CreEns) [Nguyen et al., 2025], Credal Bayesian Deep Learning (CreBNN) [Caprio et al., 2023], and Credal Deep Ensembles (CreNet) [Wang et al., 2024a]. These methods represent the current state-of-art for credal prediction and to the best of our knowledge we are the first to do a systematic comparison of these approaches in a unified benchmark. Since coverage and efficiency are not well-defined for Bayesian methods, a direct comparison to such methods is not possible.

The code for all implementations and experiments is published in a Github repository[1]. Further experimental details of our method and the implementation of the baselines are provided in Appendix B.

### 5.1 Predictive Performance

We use our method to train an ensemble of fully-connected neural networks on the embeddings of the ChaosNLI dataset [Nie et al., 2020]. This dataset features premise-hypothesis pairs in textual form

---

[1]https://github.com/timoverse/credal-prediction-relative-likelihood

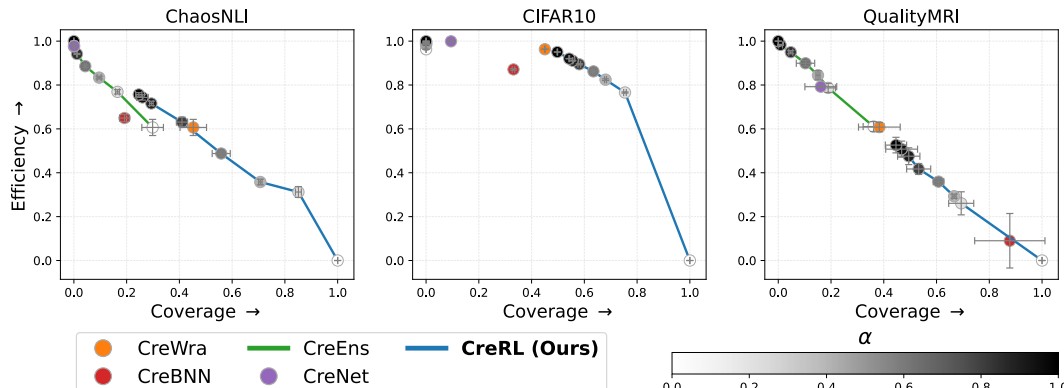

**Figure 4: Pareto front between coverage and efficiency** for ChaosNLI, CIFAR-10, and QualityMRI. CreRL (Ours) and CreEns allow trading off efficiency and coverage by varying $\alpha$. The other baselines do not allow direct adaptation of the credal set and hence result in a single point.

with the goal to classify each instance as one of three classes: entailed, neutral, and contradicted. Each instance in ChaosNLI has 100 annotations, which can be seen as a ground-truth distribution. We train an ensemble of ResNet18 models [He et al., 2016] based on our method on the CIFAR-10 dataset [Krizhevsky et al., 2009]. This dataset consists of 10 classes of images, commonly used as a benchmark for image classification tasks. We use the human annotations provided by the CIFAR-10H dataset [Peterson et al., 2019] as ground-truth distributions. In addition, we train an ensemble of ResNet18 models on the QualityMRI dataset from Schmarje et al. [2022]. This dataset comprises MRI images with multiple annotations from radiologists, each providing a quality assessment of the image [Obuchowicz et al., 2020]. These multi-class annotations were merged into annotations for two classes [Schmarje et al., 2022]. The uncertainty in these labels, as represented by the high average entropy reported in Appendix C, confirms the importance of uncertainty quantification in medical settings [Löhr et al., 2024].

To train the neural networks, the annotations are converted into a probability distribution, which we consider to be the ground-truth. The neural networks are then trained on targets that are sampled from the ground-truth probability distribution. Thus, our method does not require ground-truth probability distributions, or soft labels, for training.

We evaluate our method using the coverage and efficiency as defined in (2) and (3), where the expectation is approximated by averaging over the instances in the test set of the respective dataset. The results can be found in Figure 4. Appendix C, we report the accuracy of the ensemble members for each implementation.

**Trade-off between Efficiency and Coverage.** In general, both our method (CreRL$_\alpha$) and the CreEns$_\alpha$ approach are flexible and allow to trade off efficiency and coverage of the credal predictor by varying $\alpha$. However, CreEns can only attain credal sets in the low coverage region, whereas our approach can predict credal sets with a wide range of coverage values, including high coverage. The other methods only deliver fixed credal sets, meaning there is no direct way to to increase (or decrease) the coverage and efficiency. The CreWra approach and CreEns$_0$ (with $\alpha = 0$) will generally have the same efficiency, but a different coverage, due to the different credal set constructions (interval vs. convex hull). The CreNet converges to a point prediction, because its loss function is constructed such that the lower probability converges to the upper probability, and hence is generally in the top left region of the Pareto figure. We refer to Appendix A for more details about these methods.

For the ChaosNLI dataset, our approach dominates the CreBNN approach and the CreWra resembles a point on the Pareto front obtained by our method. As mentioned, the CreEns$_\alpha$ method can only reach the low coverage region and the CreNet is in the top left.

On CIFAR-10, our method generates a Pareto front in a region of high efficiency and high coverage. The CreWra baseline is close to our Pareto front. Since all ensemble members have high performance, as illustrated by their accuracies in Appendix C, and there is no explicit diversity promotion in

**Table 1: Out-of-Distribution detection based on epistemic uncertainty.** CIFAR-10 is used as the in-Distribution dataset. The mean and standard deviation over 3 runs are reported. Best performance is in **bold**.

| Method | SVHN | Places | CIFAR-100 | FMNIST | ImageNet |
|--------|------|--------|-----------|--------|----------|
| CreWra | **0.957**±**0.003** | 0.916±0.001 | **0.916**±**0.000** | 0.952±0.000 | **0.890**±**0.001** |
| CreEns$_{0.0}$ | 0.955±0.001 | 0.913±0.000 | 0.914±0.001 | 0.949±0.001 | 0.888±0.000 |
| CreBNN | 0.907±0.006 | 0.885±0.002 | 0.880±0.002 | 0.935±0.002 | 0.859±0.002 |
| CreNet | 0.943±0.003 | **0.918**±**0.000** | 0.912±0.000 | 0.951±0.002 | 0.884±0.001 |
| CreRL$_{1.0}$ | 0.948±0.003 | **0.918**±**0.002** | **0.916**±**0.001** | **0.957**±**0.002** | **0.889**±**0.002** |
| CreRL$_{0.95}$ | 0.917±0.013 | 0.910±0.001 | 0.901±0.000 | 0.945±0.004 | 0.878±0.002 |
| CreRL$_{0.9}$ | 0.918±0.011 | 0.907±0.001 | 0.896±0.001 | 0.944±0.004 | 0.874±0.001 |
| CreRL$_{0.8}$ | 0.906±0.008 | 0.894±0.001 | 0.884±0.003 | 0.936±0.009 | 0.865±0.002 |
| CreRL$_{0.6}$ | 0.862±0.035 | 0.874±0.003 | 0.852±0.002 | 0.893±0.005 | 0.837±0.003 |
| CreRL$_{0.4}$ | 0.739±0.029 | 0.821±0.007 | 0.796±0.007 | 0.815±0.020 | 0.788±0.010 |
| CreRL$_{0.2}$ | 0.582±0.041 | 0.736±0.010 | 0.700±0.013 | 0.676±0.046 | 0.698±0.013 |

CreEns$_\alpha$, it produces small sets for every $\alpha$. CreNet, again, produces a (close to) point prediction with low coverage. We refer to Appendix B for more details.

For QualityMRI, the story is largely consistent with that of the ChaosNLI dataset. The difference is that our method is only positioned in the high coverage half of the Pareto figure.

The ability of our method to target the high coverage area can be explained by the fact that we promote diversity, by means of our ToBias initialization, in order to promote high coverage, at the cost of efficiency. In addition, one may argue that the low coverage region represents credal sets that are of limited benefit for practitioners, as they are unlikely to demand a coverage of, say, 0.3.

Furthermore, Figure 4 shows that the shape and position of the Pareto front varies across datasets suggesting that the coverage-efficiency trade-off is inherently data-dependent. In particular, the better the maximum likelihood model, the more the Pareto front should move towards the top right (high coverage, high efficiency), which we also observe in the case of CIFAR-10.

To further study the behavior of our method, we present ablation experiments in Appendix C, varying the number of members in the ensemble and the ToBias initialization constant.

## 5.2 Out-of-Distribution Detection

Due to a lack of ground-truth uncertainties for most datasets, uncertainty methods are usually evaluated on downstream tasks such as OoD detection. We perform the OoD task to assess the (epistemic) uncertainty representation of our method. The predictor is trained on a dataset called the in-Distribution (iD) dataset and at test time introduced to instances from both the iD dataset and another OoD dataset that it has not seen before. An effective epistemic uncertainty representation should assign higher epistemic uncertainty to OoD instances than to iD instances. In the literature, this is commonly evaluated by computing epistemic uncertainty and using it to distinguish between iD and OoD instances. We adopt the epistemic uncertainty measure based on the additive decomposition proposed (and axiomatically justified) by Abellán et al. [2006]:

$$\text{EU}(\hat{\mathcal{Q}}_{\boldsymbol{x},\alpha}) = \overline{S}(\hat{\mathcal{Q}}_{\boldsymbol{x},\alpha}) - \underline{S}(\hat{\mathcal{Q}}_{\boldsymbol{x},\alpha}), \tag{5}$$

where $\overline{S}(\hat{\mathcal{Q}}_{\boldsymbol{x},\alpha})$ denotes the maximum Shannon entropy of all distributions in $\hat{\mathcal{Q}}_{\boldsymbol{x},\alpha}$ and, likewise, $\underline{S}(\hat{\mathcal{Q}}_{\boldsymbol{x},\alpha})$ the minimum entropy.[2] The performance is then measured using the AUROC. Appendix B.4 provides details about the implementation of the optimization.

We utilize the same CIFAR-10 ensembles of ResNet18 models and compare to the same baselines as in Section 5.1. We introduce SVHN Netzer et al. [2011], Places365 Zhou et al. [2018], CIFAR-100 Krizhevsky et al. [2009], FashionMNIST Xiao et al. [2017], and ImageNet Deng et al. [2009] as out-of-distribution datasets, and compute epistemic uncertainty as in (5). Our method (CreRL$_\alpha$) and

---

[2]The bounds of Shannon entropy are computed numerically using SciPy [Virtanen et al., 2020].

the Credal Ensembling (CreEns$_\alpha$) approach make use of a hyper-parameter $\alpha$. The results with the best performing $\alpha$ are presented here. The results, summarized in Table 1, present the OoD detection performance. We provide further results for both methods with different $\alpha$ values in Appendix C.

Our method (CreRL$_\alpha$) with $\alpha = 1$ performs either the best or is on par with the best methods in OoD detection, except for the SVHN dataset, where the CreWra approach has a small advantage. This shows the good (epistemic) uncertainty representation of our method.

**Influence of $\alpha$.** Furthermore, the value of the $\alpha$ parameter of our method has a significant impact on the behavior of credal sets for both iD and OoD data. We expect greater epistemic uncertainty for instances from a distribution that differs from the training data, as predictions may diverge for these instances. When $\alpha$ is close to 1, the resulting set consists of models with high relative likelihood, leading to smaller credal sets and thus lower epistemic uncertainty for iD data. Conversely, a smaller $\alpha$ includes low relative likelihood models, resulting in higher epistemic uncertainty. Given that $\alpha$ strongly influences epistemic uncertainty for iD instances, we expect a larger $\alpha$ to increase the discrepancy in uncertainty between iD and OoD instances. As shown in Table 1, across all OoD datasets, a larger $\alpha$ indeed leads to higher AUROC scores, confirming that larger $\alpha$ values lead to better separation of iD and OoD instances.

## 6  Conclusion

We proposed a new approach to generating credal predictors in a machine learning setting with complex learners on the basis of relative likelihood, which is used to identify a set of "plausible" models. The relative likelihood is an intuitive notion of model plausibility, grounded in classical statistics. Given a query instance for which a prediction is sought, a credal set is obtained as the collection of probability distributions predicted by all plausible models. Specifically, we introduced a novel method for training neural networks to approximate the credal set generated by an $\alpha$-cut of models. In order to obtain a good approximation, the $\alpha$-cut needs to be covered well, i.e., the included hypotheses should be sufficiently spread over the hypothesis space. To this end, we proposed ToBias initialization as a simple but effective diversification strategy. Furthermore, the parameter $\alpha$, which specifies the likelihood ratio between the maximum likelihood model and the "weakest" model that is still included in the $\alpha$-cut, can be adjusted depending on the task at hand and used to control the trade-off between the coverage and efficiency of credal predictions.

Experimentally, we have shown that this workflow provides strong coverage of the ground-truth conditional distributions, while maintaining a competitive efficiency, as well as performing competitively in downstream tasks such as OoD detection. The relation between $\alpha$ and the performance on different tasks sheds further light on the trade-off between diversity and prediction performance.

**Limitations and Future Work.** Our method has so far been evaluated exclusively on neural networks. However, the general concept of relative likelihood is applicable to any model class. Certain aspects of our method — such as ToBias initialization — are specifically tailored to neural networks and would have to be adapted to other learners. A promising direction for future work would be to extend this framework to other model classes.

When generating a convex credal set from our finite approximation, we include all probability distributions such that the respective class probabilities are bounded by their lower and upper probabilities. This makes the optimization problem of quantifying uncertainty easier, but it also generates sets that are larger than necessary. An alternative would be to take the convex hull of the probability distributions as the credal set, which would generate more efficient sets. Future work may explore the effect of this on metrics such as coverage or OoD performance.

Another interesting question that arises relates to the approximation quality of our approach. Having specified a threshold for the relative likelihood, there exists some ground-truth credal set for an instance. It could be interesting to explore how many probability distributions, and thus predictors, are required to accurately approximate this credal set.

**Broader Impacts.** This work contributes to the development of reliable machine learning models by improving uncertainty quantification. We do not foresee any direct negative broader impacts arising from this.

## Acknowledgments and Disclosure of Funding

We gratefully thank the anonymous reviewers for their valuable feedback, which helped improve this work. This work has received funding from the European Union's Horizon Europe research and innovation programme under the Marie Sklodowska-Curie grant agreement No 101073307. Felix Mohr participated through the project ING-312-2023 at Universidad de La Sabana. We acknowledge the Munich Center for Machine Learning (MCML) for their support of this work.

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

# A  Extended Related Work

**Uncertainty Representation.** In machine learning, uncertainty is often represented by Bayesian methods [Mackay, 1992, Blundell et al., 2015]. Frequently, the Bayesian posterior is approximated by ensembles [Lakshminarayanan et al., 2017] or methods such as Dropout [Gal and Ghahramani, 2016] or Laplace approximation [Daxberger et al., 2021]. An important characteristic of such representations, especially for uncertainty tasks, is diversity [D'Angelo and Fortuin, 2021, Wood et al., 2023]. Some works enforce this by means of regularization [de Mathelin et al., 2023], whereas others vary hyperparameters across models within the ensemble [Wenzel et al., 2020] or enforce diversity in the representations [Lopes et al., 2022].

Alternatively, credal sets have been used in the fields of imprecise probability and machine learning to represent model uncertainty [Zaffalon, 2001, Corani and Zaffalon, 2008, Corani and Mignatti, 2015]. Antonucci et al. [2012a] proposed to generate such sets based on relative likelihoods. Relative likelihoods, also referred to as normalized likelihoods, have also been used in machine learning with simple model classes such as logistic regression [Senge et al., 2014, Cella and Martin, 2024]. In this work, we build upon these approaches and address the challenges that emerge when adapting the relative likelihood to a setting with complex predictors.

Recently, credal sets have been applied in the context of machine learning. Wang et al. [2024b] take multiple samples from a Bayesian posterior or ensembles and derive class-wise lower and upper probabilities. Based on these samples, a credal set is constructed by including all probability distributions such that the individual predicted class probabilities are in the respective lower to upper class probability interval. Nguyen et al. [2025] also construct credal sets from ensemble predictions, but with the additional option of discarding potential outliers to prevent the set from becoming too large. This is done by comparing all ensemble predictions to a representative prediction, e.g. the mean prediction, using some distance between distributions, and keeping only $(1 - \alpha) \cdot 100\%$ closest predictions. The credal set is then constructed by taking the convex hull of the remaining probability distributions. Other methods directly train neural networks to predict intervals by explicitly predicting a lower and upper probability for every class [Wang et al., 2024a]. In combination with a custom loss function, consisting of regular cross-entropy for the upper probabilities and cross-entropy computed on the highest loss subset of a batch for the lower probabilities, the claim is that this approach encourages both "optimistic" and "pessimistic" predictions. Credal sets are constructed by taking the same interval-based approach as [Wang et al., 2024b]. Besides this, hybrid methods, combining multiple uncertainty frameworks, have also been proposed. Caprio et al. [2023] combine Bayesian deep learning and credal sets by considering sets of priors over weights of neural networks. Training these neural network by variational inference then results in a set of posteriors. Based on this, a credal set is constructed by sampling from the Bayesian neural networks and taking the convex hull of the sampled probability distributions. Another approach leverages conformal prediction to construct credal sets with validity guarantees [Javanmardi et al., 2024]. However, this work uses distributions over classes to perform the conformal calibration step. Since our method does not require such data, we exclude this method as a baseline. Our work distinguishes itself from aforementioned works by using relative likelihood cuts which allow for an intuitive and adaptable construction of the credal set.

Moreover, our approach is conceptually related to Rashomon sets [Semenova et al., 2022]. Both characterize a collection of models whose performance exceeds a given threshold, thereby facing the same challenge in approximating this set of plausible models [Donnelly et al., 2025]. However, the objectives differ: Rashomon sets are primarily concerned with interpretability and (syntactic) model diversity, whereas our focus is on uncertainty quantification and predictive diversity.

**Uncertainty Quantification.** Given a credal representation of uncertainty, there are many ways to quantify uncertainty. Some measures consider only the epistemic uncertainty [Abellán and Moral, 2000], whereas others use entropy to reason about the total uncertainty [Abellán and Moral, 2003]. In this line of work, Abellán et al. [2006] also proposed measures based on entropy that decompose the total uncertainty into an aleatoric and epistemic component. Conversely, Antonucci et al. [2012b] quantify uncertainty by measuring the lack of dominance of one class over others in the predictive distributions represented by the credal set. Hüllermeier et al. [2022] offer a critical analysis of these measures and propose an alternative for the dominance-based measure. Recently, Hofman et al. [2024] proposed to quantify credal uncertainty based on a decomposition of scoring rules. In the following, we consider the uncertainty measures by Abellán et al. [2006] in order to ensure a fair comparison with previous works.

# B  Experimental Details

## B.1  Datasets

**ChaosNLI**  The ChaosNLI dataset, introduced by Nie et al. [2020], is a large-scale dataset designed to study human disagreement in natural language inference (NLI) tasks. It comprises 100 human annotations per example for 3,113 examples from the SNLI and MNLI datasets, and 1,532 examples from the $\alpha$NLI dataset, totaling approximately 464,500 annotations. In line with Javanmardi et al. [2024], we use only the SNLI and MNLI subsets of the ChaosNLI dataset, but for simplicity, we will refer to this as the ChaosNLI dataset. Each example includes metadata such as the unique identifier, counts of each label assigned by annotators, the majority label, label distribution, entropy of the label distribution, the original example text, and the original label from the source dataset. This dataset enables a detailed analysis of the distribution of human opinions in NLI tasks, highlighting instances of high disagreement and questioning the validity of using majority labels as the sole ground truth. ChaosNLI is publicly available under the Creative Commons Attribution-NonCommercial 4.0 International (CC BY-NC 4.0) license. We train our models on the 768-dimensional embeddings of the ChaosNLI dataset retrieved from `https://github.com/alireza-javanmardi/conformal-credal-sets`. We refer to [Javanmardi et al., 2024] for more details on the generation of the embeddings.

**CIFAR-10**  The CIFAR-10 dataset is a widely used benchmark in machine learning and computer vision, introduced by Krizhevsky et al. [2009], and Geoffrey Hinton in 2009. It comprises 60,000 color images at a resolution of 32×32 pixels, evenly distributed across 10 distinct classes: airplane, automobile, bird, cat, deer, dog, frog, horse, ship, and truck. The dataset is partitioned into 50,000 training images and 10,000 test images, organized into five training batches and one test batch, each containing 10,000 images. The dataset is publicly available and has been utilized extensively for developing and benchmarking machine learning models. While the original dataset does not specify a license, various distributions, such as those provided by TensorFlow Datasets, are released under the Creative Commons Attribution 4.0 License.

**CIFAR-10H**  The CIFAR-10H dataset provides human-derived soft labels for the 10,000 images in the CIFAR-10 test set, capturing the variability in human annotation during image classification tasks. Developed by Peterson et al. [2019], the dataset comprises 511,400 annotations collected from 2,571 Amazon Mechanical Turk workers, with each image receiving approximately 51 labels. Annotators classified images into one of the ten CIFAR-10 categories, enabling the construction of probability distributions over labels for each image. CIFAR-10H is publicly available under the Creative Commons BY-NC-SA 4.0 license.

**CIFAR-100**  The CIFAR-100 dataset, introduced by Krizhevsky et al. [2009], comprises 60,000 color images at 32×32 resolution, divided into 100 classes with 600 images each. Each image has a "fine" label (specific class) and a "coarse" label (superclass), with the 100 classes grouped into 20 superclasses. The dataset is split into 50,000 training and 10,000 test images. It is a subset of the Tiny Images dataset and is commonly used for evaluating image classification algorithms. While the original dataset does not specify a license, various distributions, such as those provided by TensorFlow Datasets, are released under the Creative Commons Attribution 4.0 License.

**QualityMRI**  The QualityMRI dataset, introduced by Obuchowicz et al. [2020], is part of the Data-Centric Image Classification (DCIC) Benchmark, which aims to evaluate the impact of dataset curation on model performance. The dataset contains 310 magnetic resonance (MRI) images spanning various quality levels and is designed to assess the MRI image quality. The dataset is publicly available under the Creative Commons BY-SA 4.0 license.

**SVHN**  The SVHN dataset, introduced by Netzer et al. [2011], consists of over 600,000 32×32 RGB images of digits (0–9) obtained from real-world house number images in Google Street View. It includes three subsets: 73,257 training images, 26,032 test images, and 531,131 additional images for extra training. The dataset is designed for digit recognition tasks with minimal preprocessing. While the original dataset does not specify a license, various distributions, such as those provided by TensorFlow Datasets, are released under the Creative Commons Attribution 4.0 License.

**Places365** Places365, introduced by Zhou et al. [2018], is a large-scale scene recognition dataset containing 1.8 million training images across 365 scene categories. The validation set includes 50 images per category, and the test set has 900 images per category. An extended version, Places365-Challenge-2016, adds 6.2 million images and 69 new scene classes, totaling 8 million images over 434 categories. While the original dataset does not specify a license, various distributions, such as those provided by TensorFlow Datasets, are released under the Creative Commons Attribution 4.0 License.

**FMNIST** Fashion-MNIST (FMNIST), introduced by Xiao et al. [2017], is a dataset of Zalando's article images, comprising 70,000 28×28 grayscale images labeled across 10 classes, such as T-shirt/top, Trouser, and Sneaker. It includes 60,000 training and 10,000 test images and serves as a direct replacement for the original MNIST dataset for benchmarking machine learning algorithms. FMNIST is publicly available under the MIT License.

**ImageNet** ImageNet, introduced by Deng et al. [2009], is a large-scale image database organized according to the WordNet hierarchy, containing over 14 million images across more than 20,000 categories. The ILSVRC subset (ImageNet-1K) includes 1,281,167 training images, 50,000 validation images, and 100,000 test images across 1,000 classes. The dataset is available for free to researchers for non-commercial use.

## B.2 Models

**Fully-Connected Network** We train fully-connected neural networks on the ChaosNLI dataset. The network consists of 4 linear layers with $[768 - 256 - 64 - 16 - 3]$ units with ReLU activations, except for the last layer, which has Softmax transformation to transform the logits into probabilities. We use the hyperparameters (cf. Table 2) similar to the optimal parameters found in Javanmardi et al. [2024].

**ResNet18** For experiments on the CIFAR-10 dataset, we use the PyTorch ResNet-18 implementation and hyperparameters provided by `https://github.com/kuangliu/pytorch-cifar`. This model is specifically optimized for CIFAR-10 and is trained from scratch, without any pretraining on ImageNet. For experiments on the QualityMRI dataset, we use the ResNet18 implementation from the PyTorch torchvision package with random initialization, i.e. no pretrained weights.

**Hyperparameters** Each dataset is trained using a dedicated set of hyperparameters as presented in Table 2. We evaluated multiple configurations and selected the best-performing ones for each dataset. To ensure fair and consistent comparisons, all models trained on a given dataset, both our approach and the baselines, use the same hyperparameter settings. The only exception is the CreBNN, which requires a KL-divergence penalty of $1e - 7$ and zero weight decay when using the Adam optimizer [Kingma and Ba, 2015]. When we apply the SGD optimizer with a learning rate scheduler, namely Cosine Annealing [Loshchilov and Hutter, 2017], CreBNN requires additionally a momentum of $0.9$ to enable effective learning.

**Table 2: Hyperparameters used for each dataset.**

| Hyperparameter | ChaosNLI | CIFAR-10 | QualityMRI |
|---|---|---|---|
| Model | FCNet | ResNet18 | ResNet18 |
| Epochs | 300 | 200 | 200 |
| Learning rate | 0.01 | 0.1 | 0.01 |
| Weight decay | 0.0 | 0.0005 | 0.0005 |
| Optimizer | Adam | SGD | SGD |
| Ensemble members | 20 | 20 | 20 |
| LR scheduler | - | CosineAnnealing | CosineAnnealing |
| Tobias value | 100 | 100 | 100 |

### B.3 Out-of-Distribution Detection

We use the SVHN [Netzer et al., 2011], Places365 [Zhou et al., 2018], CIFAR-100 [Krizhevsky et al., 2009], Fashion-MNIST [Xiao et al., 2017], and ImageNet [Le and Yang, 2015] datasets for Out-of-Distribution detection.

The Out-of-Distribution task is treated as a binary classification task where the epistemic uncertainty is used as the classification criterion. In order to balance the data, we sample 10000 instances from the test set of the respective datasets. On both the in-Distribution data (CIFAR-10) and the Out-of-Distribution data, the same transforms — normalization and resizing to 32 by 32 pixels — are applied. After computing the epistemic uncertainty the area under the receiver operating characteristics curve (AUROC) is computed and used as the comparison metric.

### B.4 Computing Uncertainty

Computing the total and aleatoric uncertainty involves solving a per instance optimization problem, after which the epistemic uncertainty is obtained as their difference (see Equation (5)). This optimization is performed using SciPy's `minimize` function with the SLSQP solver and the default parameters.

**Interval-Based**    For the interval-based credal sets, the optimization is initialized with the mean of the predicted distributions, bounded between the lower and upper probabilities for each class, and constrained to ensure that the solution forms a valid probability distribution, namely, the class probabilities must sum to 1.

**Convex Hull**    For the credal sets based on the convex hull, the optimization is done on the weights of convex combination, instead of the probability distribution. Uniform weights are used as the initial value, the weights are bounded between 0 and 1, and constrained to sum to 1.

**Estimated Computing Time**    Here, we provide an estimated upper bound of the computation time for computing the lower and upper entropy of a credal set based on the example of the largest possible credal set, namely the full probability simplex for a 10-class problem, such as CIFAR-10. For this largest set, computing the upper entropy using an interval-based credal set takes on average 0.03 seconds, while computing the lower entropy takes about 0.02 seconds. For the convex hull-based credal set of the same size, the average computation time is 0.07 seconds for upper entropy and 0.03 seconds for lower entropy. To simplify our estimation, we average these times and assume that for a single instance it takes 0.03 seconds to optimize the lower entropy and 0.04 seconds to optimize upper entropy, summing up to 0.07 seconds of computing time per instance.

In the Out-of-Distribution (OoD) detection experiments, we need to compute both lower and upper entropy for each instance in both the in-Distribution (iD) and OoD datasets. Each dataset contains $10,000$ instances, resulting in roughly 23 minutes of computation time per model to obtain the epistemic uncertainty across all instances. Since we run three seeds per model and evaluate 12 different models, the total runtime for a single OoD experiment on one dataset is approximately 14 hours. As we evaluate OoD detection across five datasets, the total computing time amounts to around 70 hours excluding any time needed to train models beforehand.

### B.5 Computing Coverage

Coverage is evaluated by checking whether the ground-truth distribution lies within the predicted credal set. For the interval-based approach, this involves verifying that each class probability of the ground-truth distribution falls between the corresponding lower and upper bounds of the credal set. For the convex hull approach, we assess whether the ground-truth distribution can be expressed as a convex combination of the extreme points defining the credal set. This optimization is done using the SciPy `linprog` function.

### B.6 Baselines

We list all details regarding the implementations of the baselines that were used in the paper. In general, all baselines were implemented in our own code base.

**Credal Wrapper (CreWra)**   The Credal Wrapper was initially implemented in TensorFlow, but we reimplemented it in PyTorch to ensure compatibility with our framework. It follows a standard ensemble learning approach, training multiple models independently. Like our method, the Credal Wrapper constructs credal sets using class-wise upper and lower probability bounds, making it well-aligned with our implementation. Overall, we closely follow both the original paper and their available implementation [Wang et al., 2024b].

**Credal Ensembling (CreEns$_\alpha$)**   Since no official code was available for neural network implementations of Credal Ensembling, we reimplemented the method ourselves. Our implementation closely follows all details provided in Nguyen et al. [2025]. The approach builds on standard ensemble training procedures, with inference adapted according to their proposed method of sorting predictions based on a distance measure and selecting only $\alpha\%$ closest predictions to construct credal sets. In our experiments, we use the Euclidean distance measure and evaluate several values of $\alpha$.

**Credal Deep Ensembles (CreNet)**   As the official implementation of Credal Deep Ensembles is only available in TensorFlow, we reimplemented the method in PyTorch to ensure compatibility with our codebase. Our implementation closely mirrors the original TensorFlow code, particularly in adapting the model architecture and loss function. Specifically, we replace each model's final linear layer with a head comprising a linear layer outputting $2 \times$ classes (representing upper and lower probability bounds), followed by a batch normalization layer and the custom IntSoftmax layer. We also reimplemented the proposed loss function, which computes a cross-entropy loss for the upper bounds and selectively backpropagates the lower-bound loss only for the $\delta\%$ of samples with the highest loss values, as described in Wang et al. [2024a]. In our experiments, we use $\delta = 0.5$, as suggested in Wang et al. [2024a].

**Credal Bayesian Deep Learning (CreBNN)**   No code or implementation details for Credal Bayesian Deep Learning (CreBNN) were made publicly available, and despite multiple attempts to contact the authors, we received no response or further clarification. As a result, we reimplemented the method ourselves based solely on the high-level description provided in the paper. In our implementation, each ensemble member is a Bayesian neural network (BNN) trained with variational inference using different priors, with prior means $\mu$ sampled from $[-1, 1]$ and standard deviations $\sigma$ from $[0.1, 2]$ to form a diverse prior set. During inference, we draw one sample from each BNN to obtain a finite set of probability distributions, and construct the credal set as the convex hull of these predictions.

### B.7   Computing Resources

To run the experiments presented in this work, we utilized the computing resources detailed in Table 3. The total estimated GPU usage amounts to approximately 750 hours.

**Table 3: Specifications of Computing Resources.**

| Component | Specification |
| --- | --- |
| CPU | AMD EPYC MILAN 7413 Processor, 24C/48T 2.65GHz 128MB L3 Cache |
| GPU | 2 × NVIDIA A40 (48 GB GDDR each) |
| RAM | 128 GB (4x 32GB) DDR4-3200MHz ECC DIMM |
| Storage | 2 × 480GB Samsung Datacenter SSD PM893 |

# C   Additional Experiments

This section presents additional results, including ablation studies and alternative hyperparameter configurations, that complement the findings reported in the main paper.

## C.1   Out-of-Distribution detection

In addition to the OoD experiments in Section 5, we evaluate a broad range of values $\alpha$ for the CreEns approach.

**Table 4: Out-of-Distribution detection based on epistemic uncertainty.** CIFAR-10 is used as the in-Distribution dataset. The mean and standard deviation over 3 runs are reported. Best performance is in **bold**.

| Method | SVHN | Places | CIFAR-100 | FMNIST | ImageNet |
|---|---|---|---|---|---|
| CreWra | **0.957**±0.003 | 0.916±0.001 | **0.916**±0.000 | 0.952±0.000 | **0.890**±0.001 |
| CreEns$_{0.95}$ | 0.500±0.000 | 0.500±0.000 | 0.500±0.000 | 0.500±0.000 | 0.500±0.000 |
| CreEns$_{0.9}$ | 0.921±0.002 | 0.879±0.002 | 0.883±0.001 | 0.915±0.001 | 0.857±0.002 |
| CreEns$_{0.8}$ | 0.937±0.001 | 0.896±0.001 | 0.900±0.001 | 0.929±0.001 | 0.875±0.001 |
| CreEns$_{0.6}$ | 0.944±0.002 | 0.902±0.001 | 0.906±0.000 | 0.935±0.001 | 0.881±0.001 |
| CreEns$_{0.4}$ | 0.947±0.001 | 0.906±0.001 | 0.908±0.000 | 0.940±0.001 | 0.883±0.001 |
| CreEns$_{0.2}$ | 0.950±0.001 | 0.909±0.001 | 0.911±0.000 | 0.946±0.001 | 0.885±0.001 |
| CreEns$_{0.0}$ | 0.955±0.001 | 0.913±0.000 | 0.914±0.001 | 0.949±0.001 | 0.888±0.000 |
| CreNet | 0.943±0.003 | 0.918±0.000 | 0.912±0.000 | 0.951±0.002 | 0.884±0.001 |
| CreBNN | 0.907±0.006 | 0.885±0.002 | 0.880±0.002 | 0.935±0.002 | 0.859±0.002 |
| CreRL$_{1.0}$ | 0.948±0.003 | **0.918**±0.002 | **0.916**±0.001 | **0.957**±0.002 | **0.889**±0.002 |
| CreRL$_{0.95}$ | 0.917±0.013 | 0.910±0.001 | 0.901±0.000 | 0.945±0.004 | 0.878±0.002 |
| CreRL$_{0.9}$ | 0.918±0.011 | 0.907±0.001 | 0.896±0.001 | 0.944±0.004 | 0.874±0.001 |
| CreRL$_{0.8}$ | 0.906±0.008 | 0.894±0.001 | 0.884±0.003 | 0.936±0.009 | 0.865±0.002 |
| CreRL$_{0.6}$ | 0.862±0.035 | 0.874±0.003 | 0.852±0.002 | 0.893±0.005 | 0.837±0.003 |
| CreRL$_{0.4}$ | 0.739±0.029 | 0.821±0.007 | 0.796±0.007 | 0.815±0.020 | 0.788±0.010 |
| CreRL$_{0.2}$ | 0.582±0.041 | 0.736±0.010 | 0.700±0.013 | 0.676±0.046 | 0.698±0.013 |

## C.2   Ablations

In the following ablation study, we evaluate two factors: the impact of varying the initialization constant in our proposed ToBias initialization, and the effect of changing the number of ensemble members. The study is conducted on the ChaosNLI dataset.

### C.2.1   ToBias Initialization

We study the impact of the ToBias initialization constant $\beta$ on the coverage and efficiency of the resulting credal predictor on the ChaosNLI dataset. To do so, we vary the constant, taking values $\beta \in \{5, 10, 20, 30, 50, 80, 100, 200, 500\}$. The ensemble is then trained as proposed in Algorithm 1. The coverage and efficiency Pareto front is shown in Figure 5 with the mean and standard deviation over three runs. We split the results into three Figures for better readability.

For low values of $\beta$, the coverage of the ensembles is rather low and as $\beta$ increases, the coverage also increases. For large values of $\beta$, e.g. $\beta = 200$ and $\beta = 500$, the ensembles with large $\alpha$ values are no longer able to reach into the low coverage, high efficiency region. In particular, $\beta = 500$ has a higher coverage and lower efficiency for larger values of $\alpha$ than for smaller values of $\alpha$. This may caused by converge problems due to having a very large bias for a particular class. This causes the individual ensemble members to not always converge to their desired threshold, hence resulting in large credal sets (with high coverage and low efficiency), because the border of the credal set is not reached. In essence, the $\beta$ value provides the ability to slightly shift the Pareto front to reach the desired coverage, efficiency region (in addition to $\alpha$).

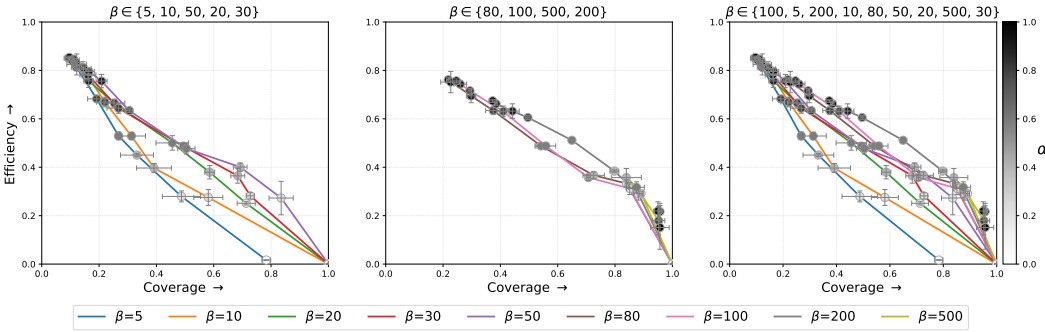

**Figure 5: Pareto front between coverage and efficiency** for different values of ToBias constant $\beta$. **Left:** low values of $\beta$, **middle:** high values of $\beta$, **right:** all values of $\beta$.

### C.2.2 Number of Ensemble Members

We study the impact of the numbers of ensemble members $M$ on the coverage and efficiency of the resulting credal predictor on the ChaosNLI dataset. To do so, we vary the constant, taking values $M \in \{1, 2, 3, 4, 5, 10, 20, 30, 50\}$. The ensemble is then trained as proposed in Algorithm 1. The coverage and efficiency Pareto front is shown in Figure 6 with the mean and standard deviation over three runs. We split the results into three Figures for better readability.

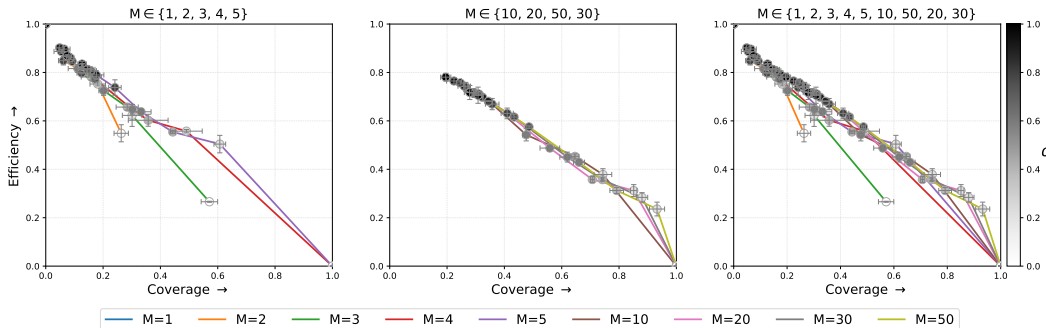

**Figure 6: Pareto front between coverage and efficiency** for different number of ensemble members $M$. **Left:** low values of $M$, **middle:** high values of $M$, **right:** all values of $M$.

Naturally, when $M = 1$, the coverage is 0 and efficiency 1, because the credal set reduces to a point prediction. With $M = 2$ and $M = 3$, the $\alpha = 0$ ensemble does not have coverage 1 and efficiency 0, because there are not enough ensemble members, and hence probability distribution, to span the whole probability simplex for this 3 class problem. As $M$ increases, the Pareto front starts to span more of the Pareto figure, until stabilizing around $M = 20$. After this, the number of ensemble members does not have a significant impact on the resulting Pareto front anymore.

### C.3 Data Uncertainty

The datasets used in our experiments, namely ChaosNLI, CIFAR-10, and QualityMRI, differ in their inherent levels of (aleatoric) uncertainty. This is reflected, for example, in the degree of disagreement among the collected annotations, which serve as the ground-truth distributions in our evaluations. To provide further insight, we report the average entropy of these ground-truth distributions in Table 5.

**Table 5: Average entropy of ground-truth distributions** in the test sets of ChaosNLI, CIFAR-10 and QualityMRI.

| Dataset | Avg. Entropy |
|---------|--------------|
| ChaosNLI | 0.932 |
| CIFAR-10 | 0.223 |
| QualityMRI | 0.782 |

### C.4 Performance of Ensemble Members

We visualized the trade-off between coverage and efficiency in Figure 4, and provide the corresponding numerical values in Table 6. We also report the accuracies of the individual predictors within each ensemble for both the baselines and our proposed method to provide insight into their standalone performance (cf. Tables 7 and 8). For CreNet, whose ensemble members directly predict probability intervals, we use their intersection probability technique to derive pointwise predictions [Wang et al., 2024a].

**Table 6: Coverage and efficiency for different methods across datasets.**

| Approach | ChaosNLI | | CIFAR-10 | | QualityMRI | |
|---|---|---|---|---|---|---|
| | Coverage | Efficiency | Coverage | Efficiency | Coverage | Efficiency |
| $CreRL_{0.0}$ | $1.000\pm0.000$ | $0.000\pm0.000$ | $1.000\pm0.000$ | $0.000\pm0.000$ | $1.000\pm0.000$ | $0.000\pm0.000$ |
| $CreRL_{0.2}$ | $0.851\pm0.011$ | $0.312\pm0.025$ | $0.754\pm0.005$ | $0.766\pm0.003$ | $0.796\pm0.068$ | $0.261\pm0.053$ |
| $CreRL_{0.4}$ | $0.707\pm0.007$ | $0.358\pm0.013$ | $0.680\pm0.004$ | $0.824\pm0.004$ | $0.747\pm0.033$ | $0.293\pm0.010$ |
| $CreRL_{0.6}$ | $0.559\pm0.034$ | $0.488\pm0.012$ | $0.634\pm0.004$ | $0.862\pm0.004$ | $0.677\pm0.023$ | $0.360\pm0.016$ |
| $CreRL_{0.8}$ | $0.410\pm0.021$ | $0.631\pm0.022$ | $0.581\pm0.005$ | $0.894\pm0.003$ | $0.613\pm0.035$ | $0.417\pm0.025$ |
| $CreRL_{0.9}$ | $0.294\pm0.007$ | $0.716\pm0.010$ | $0.556\pm0.001$ | $0.911\pm0.001$ | $0.548\pm0.046$ | $0.475\pm0.038$ |
| $CreRL_{0.95}$ | $0.260\pm0.022$ | $0.743\pm0.012$ | $0.543\pm0.002$ | $0.920\pm0.001$ | $0.511\pm0.046$ | $0.508\pm0.037$ |
| $CreRL_{1.0}$ | $0.246\pm0.012$ | $0.757\pm0.008$ | $0.498\pm0.001$ | $0.950\pm0.000$ | $0.500\pm0.086$ | $0.526\pm0.036$ |
| CreWra | $0.453\pm0.050$ | $0.607\pm0.037$ | $0.450\pm0.001$ | $0.963\pm0.000$ | $0.355\pm0.057$ | $0.608\pm0.021$ |
| CreNet | $0.001\pm0.002$ | $0.978\pm0.008$ | $0.094\pm0.010$ | $0.999\pm0.000$ | $0.188\pm0.020$ | $0.792\pm0.003$ |
| CreBNN | $0.195\pm0.018$ | $0.649\pm0.017$ | $0.331\pm0.005$ | $0.871\pm0.003$ | $0.898\pm0.133$ | $0.090\pm0.124$ |
| $CreEns_{0.0}$ | $0.298\pm0.041$ | $0.607\pm0.037$ | $0.000\pm0.000$ | $0.963\pm0.000$ | $0.329\pm0.040$ | $0.611\pm0.023$ |
| $CreEns_{0.2}$ | $0.165\pm0.007$ | $0.769\pm0.010$ | $0.000\pm0.000$ | $0.983\pm0.000$ | $0.177\pm0.060$ | $0.787\pm0.020$ |
| $CreEns_{0.4}$ | $0.096\pm0.008$ | $0.834\pm0.006$ | $0.000\pm0.000$ | $0.986\pm0.000$ | $0.118\pm0.059$ | $0.844\pm0.021$ |
| $CreEns_{0.6}$ | $0.043\pm0.003$ | $0.886\pm0.005$ | $0.000\pm0.000$ | $0.988\pm0.000$ | $0.081\pm0.057$ | $0.899\pm0.012$ |
| $CreEns_{0.8}$ | $0.012\pm0.002$ | $0.941\pm0.002$ | $0.000\pm0.000$ | $0.993\pm0.000$ | $0.054\pm0.033$ | $0.950\pm0.008$ |
| $CreEns_{0.9}$ | $0.000\pm0.000$ | $0.976\pm0.001$ | $0.000\pm0.000$ | $0.997\pm0.000$ | $0.016\pm0.023$ | $0.983\pm0.002$ |
| $CreEns_{0.95}$ | $0.000\pm0.000$ | $1.000\pm0.000$ | $0.000\pm0.000$ | $1.000\pm0.000$ | $0.000\pm0.000$ | $1.000\pm0.000$ |

**Table 7: Accuracy per ensemble member of CreRL$_\alpha$** for varying $\alpha$ values on ChaosNLI, CIFAR-10, and QualityMRI.

| | | ChaosNLI | | | | | | | |
|---|---|---|---|---|---|---|---|---|---|
| | $\alpha$ | 0.0 | 0.2 | 0.4 | 0.6 | 0.8 | 0.9 | 0.95 | 1.0 |
| Member $m$ | 1 | $0.678_{\pm0.017}$ | $0.678_{\pm0.017}$ | $0.678_{\pm0.017}$ | $0.678_{\pm0.017}$ | $0.678_{\pm0.017}$ | $0.678_{\pm0.017}$ | $0.678_{\pm0.017}$ | $0.678_{\pm0.017}$ |
| | 2 | $0.370_{\pm0.012}$ | $0.526_{\pm0.075}$ | $0.512_{\pm0.063}$ | $0.577_{\pm0.011}$ | $0.608_{\pm0.055}$ | $0.664_{\pm0.028}$ | $0.682_{\pm0.016}$ | $0.678_{\pm0.019}$ |
| | 3 | $0.453_{\pm0.010}$ | $0.549_{\pm0.008}$ | $0.407_{\pm0.020}$ | $0.604_{\pm0.045}$ | $0.634_{\pm0.016}$ | $0.668_{\pm0.003}$ | $0.663_{\pm0.027}$ | $0.657_{\pm0.020}$ |
| | 4 | $0.178_{\pm0.014}$ | $0.468_{\pm0.022}$ | $0.483_{\pm0.045}$ | $0.483_{\pm0.046}$ | $0.651_{\pm0.016}$ | $0.660_{\pm0.012}$ | $0.661_{\pm0.010}$ | $0.667_{\pm0.006}$ |
| | 5 | $0.370_{\pm0.012}$ | $0.568_{\pm0.018}$ | $0.425_{\pm0.046}$ | $0.560_{\pm0.042}$ | $0.624_{\pm0.018}$ | $0.656_{\pm0.016}$ | $0.677_{\pm0.017}$ | $0.686_{\pm0.020}$ |
| | 6 | $0.453_{\pm0.010}$ | $0.535_{\pm0.015}$ | $0.530_{\pm0.023}$ | $0.585_{\pm0.025}$ | $0.647_{\pm0.017}$ | $0.663_{\pm0.012}$ | $0.674_{\pm0.007}$ | $0.676_{\pm0.006}$ |
| | 7 | $0.178_{\pm0.014}$ | $0.453_{\pm0.015}$ | $0.506_{\pm0.041}$ | $0.594_{\pm0.044}$ | $0.637_{\pm0.020}$ | $0.647_{\pm0.003}$ | $0.664_{\pm0.005}$ | $0.675_{\pm0.006}$ |
| | 8 | $0.370_{\pm0.012}$ | $0.585_{\pm0.027}$ | $0.494_{\pm0.119}$ | $0.585_{\pm0.039}$ | $0.646_{\pm0.038}$ | $0.673_{\pm0.022}$ | $0.677_{\pm0.024}$ | $0.682_{\pm0.014}$ |
| | 9 | $0.453_{\pm0.010}$ | $0.493_{\pm0.074}$ | $0.595_{\pm0.042}$ | $0.604_{\pm0.029}$ | $0.663_{\pm0.007}$ | $0.668_{\pm0.002}$ | $0.661_{\pm0.023}$ | $0.670_{\pm0.009}$ |
| | 10 | $0.178_{\pm0.014}$ | $0.476_{\pm0.022}$ | $0.514_{\pm0.050}$ | $0.552_{\pm0.054}$ | $0.650_{\pm0.004}$ | $0.648_{\pm0.008}$ | $0.664_{\pm0.013}$ | $0.675_{\pm0.008}$ |
| | 11 | $0.370_{\pm0.012}$ | $0.533_{\pm0.041}$ | $0.584_{\pm0.046}$ | $0.587_{\pm0.035}$ | $0.676_{\pm0.029}$ | $0.684_{\pm0.018}$ | $0.684_{\pm0.012}$ | $0.683_{\pm0.017}$ |
| | 12 | $0.453_{\pm0.010}$ | $0.644_{\pm0.011}$ | $0.638_{\pm0.015}$ | $0.650_{\pm0.011}$ | $0.659_{\pm0.011}$ | $0.670_{\pm0.013}$ | $0.672_{\pm0.005}$ | $0.665_{\pm0.019}$ |
| | 13 | $0.178_{\pm0.014}$ | $0.510_{\pm0.065}$ | $0.584_{\pm0.079}$ | $0.633_{\pm0.021}$ | $0.652_{\pm0.009}$ | $0.664_{\pm0.006}$ | $0.668_{\pm0.010}$ | $0.670_{\pm0.014}$ |
| | 14 | $0.370_{\pm0.012}$ | $0.560_{\pm0.040}$ | $0.603_{\pm0.015}$ | $0.632_{\pm0.031}$ | $0.668_{\pm0.020}$ | $0.675_{\pm0.014}$ | $0.684_{\pm0.014}$ | $0.676_{\pm0.011}$ |
| | 15 | $0.453_{\pm0.010}$ | $0.644_{\pm0.019}$ | $0.643_{\pm0.013}$ | $0.662_{\pm0.017}$ | $0.679_{\pm0.014}$ | $0.685_{\pm0.013}$ | $0.677_{\pm0.007}$ | $0.670_{\pm0.019}$ |
| | 16 | $0.178_{\pm0.014}$ | $0.630_{\pm0.021}$ | $0.637_{\pm0.001}$ | $0.647_{\pm0.013}$ | $0.651_{\pm0.005}$ | $0.668_{\pm0.017}$ | $0.658_{\pm0.007}$ | $0.671_{\pm0.002}$ |
| | 17 | $0.370_{\pm0.012}$ | $0.623_{\pm0.045}$ | $0.647_{\pm0.017}$ | $0.655_{\pm0.010}$ | $0.684_{\pm0.014}$ | $0.688_{\pm0.024}$ | $0.685_{\pm0.020}$ | $0.687_{\pm0.010}$ |
| | 18 | $0.453_{\pm0.010}$ | $0.652_{\pm0.005}$ | $0.655_{\pm0.009}$ | $0.664_{\pm0.014}$ | $0.684_{\pm0.008}$ | $0.669_{\pm0.011}$ | $0.663_{\pm0.009}$ | $0.663_{\pm0.005}$ |
| | 19 | $0.178_{\pm0.014}$ | $0.650_{\pm0.008}$ | $0.659_{\pm0.003}$ | $0.663_{\pm0.015}$ | $0.673_{\pm0.006}$ | $0.676_{\pm0.005}$ | $0.667_{\pm0.001}$ | $0.667_{\pm0.010}$ |
| | 20 | $0.370_{\pm0.012}$ | $0.689_{\pm0.017}$ | $0.677_{\pm0.014}$ | $0.679_{\pm0.024}$ | $0.684_{\pm0.029}$ | $0.688_{\pm0.024}$ | $0.671_{\pm0.026}$ | $0.673_{\pm0.033}$ |

| | | CIFAR-10 | | | | | | | |
|---|---|---|---|---|---|---|---|---|---|
| | $\alpha$ | 0.0 | 0.2 | 0.4 | 0.6 | 0.8 | 0.9 | 0.95 | 1.0 |
| Member $m$ | 1 | $0.935_{\pm0.003}$ | $0.935_{\pm0.003}$ | $0.935_{\pm0.003}$ | $0.935_{\pm0.003}$ | $0.935_{\pm0.003}$ | $0.935_{\pm0.003}$ | $0.935_{\pm0.003}$ | $0.943_{\pm0.001}$ |
| | 2 | $0.100_{\pm0.000}$ | $0.490_{\pm0.027}$ | $0.711_{\pm0.030}$ | $0.792_{\pm0.014}$ | $0.860_{\pm0.003}$ | $0.889_{\pm0.003}$ | $0.898_{\pm0.005}$ | $0.934_{\pm0.001}$ |
| | 3 | $0.100_{\pm0.000}$ | $0.582_{\pm0.043}$ | $0.725_{\pm0.027}$ | $0.809_{\pm0.004}$ | $0.873_{\pm0.008}$ | $0.891_{\pm0.002}$ | $0.898_{\pm0.004}$ | $0.934_{\pm0.002}$ |
| | 4 | $0.100_{\pm0.000}$ | $0.591_{\pm0.009}$ | $0.722_{\pm0.008}$ | $0.816_{\pm0.009}$ | $0.871_{\pm0.008}$ | $0.891_{\pm0.000}$ | $0.902_{\pm0.001}$ | $0.936_{\pm0.001}$ |
| | 5 | $0.099_{\pm0.000}$ | $0.645_{\pm0.036}$ | $0.754_{\pm0.012}$ | $0.837_{\pm0.011}$ | $0.874_{\pm0.002}$ | $0.886_{\pm0.004}$ | $0.906_{\pm0.003}$ | $0.935_{\pm0.002}$ |
| | 6 | $0.098_{\pm0.000}$ | $0.704_{\pm0.010}$ | $0.775_{\pm0.002}$ | $0.843_{\pm0.012}$ | $0.879_{\pm0.006}$ | $0.891_{\pm0.003}$ | $0.903_{\pm0.003}$ | $0.934_{\pm0.002}$ |
| | 7 | $0.100_{\pm0.000}$ | $0.697_{\pm0.012}$ | $0.782_{\pm0.004}$ | $0.851_{\pm0.008}$ | $0.881_{\pm0.003}$ | $0.893_{\pm0.006}$ | $0.902_{\pm0.002}$ | $0.936_{\pm0.001}$ |
| | 8 | $0.100_{\pm0.000}$ | $0.748_{\pm0.015}$ | $0.801_{\pm0.011}$ | $0.850_{\pm0.006}$ | $0.880_{\pm0.006}$ | $0.897_{\pm0.003}$ | $0.906_{\pm0.004}$ | $0.935_{\pm0.001}$ |
| | 9 | $0.101_{\pm0.000}$ | $0.765_{\pm0.025}$ | $0.813_{\pm0.020}$ | $0.848_{\pm0.004}$ | $0.890_{\pm0.004}$ | $0.896_{\pm0.005}$ | $0.908_{\pm0.001}$ | $0.936_{\pm0.002}$ |
| | 10 | $0.100_{\pm0.000}$ | $0.779_{\pm0.007}$ | $0.830_{\pm0.007}$ | $0.862_{\pm0.011}$ | $0.886_{\pm0.012}$ | $0.895_{\pm0.004}$ | $0.904_{\pm0.003}$ | $0.936_{\pm0.001}$ |
| | 11 | $0.100_{\pm0.000}$ | $0.790_{\pm0.003}$ | $0.831_{\pm0.003}$ | $0.865_{\pm0.005}$ | $0.889_{\pm0.007}$ | $0.904_{\pm0.001}$ | $0.908_{\pm0.004}$ | $0.936_{\pm0.001}$ |
| | 12 | $0.100_{\pm0.000}$ | $0.821_{\pm0.009}$ | $0.854_{\pm0.013}$ | $0.863_{\pm0.008}$ | $0.890_{\pm0.002}$ | $0.902_{\pm0.004}$ | $0.907_{\pm0.003}$ | $0.934_{\pm0.002}$ |
| | 13 | $0.100_{\pm0.000}$ | $0.824_{\pm0.001}$ | $0.850_{\pm0.002}$ | $0.875_{\pm0.003}$ | $0.884_{\pm0.009}$ | $0.904_{\pm0.003}$ | $0.907_{\pm0.002}$ | $0.937_{\pm0.002}$ |
| | 14 | $0.100_{\pm0.000}$ | $0.840_{\pm0.002}$ | $0.860_{\pm0.008}$ | $0.874_{\pm0.009}$ | $0.891_{\pm0.002}$ | $0.906_{\pm0.006}$ | $0.912_{\pm0.004}$ | $0.934_{\pm0.002}$ |
| | 15 | $0.099_{\pm0.000}$ | $0.849_{\pm0.007}$ | $0.868_{\pm0.006}$ | $0.879_{\pm0.004}$ | $0.898_{\pm0.003}$ | $0.905_{\pm0.001}$ | $0.911_{\pm0.000}$ | $0.933_{\pm0.002}$ |
| | 16 | $0.098_{\pm0.000}$ | $0.868_{\pm0.009}$ | $0.873_{\pm0.006}$ | $0.887_{\pm0.004}$ | $0.904_{\pm0.004}$ | $0.910_{\pm0.003}$ | $0.911_{\pm0.002}$ | $0.936_{\pm0.002}$ |
| | 17 | $0.100_{\pm0.000}$ | $0.863_{\pm0.003}$ | $0.879_{\pm0.003}$ | $0.888_{\pm0.004}$ | $0.901_{\pm0.005}$ | $0.908_{\pm0.002}$ | $0.909_{\pm0.002}$ | $0.935_{\pm0.001}$ |
| | 18 | $0.100_{\pm0.000}$ | $0.879_{\pm0.004}$ | $0.892_{\pm0.005}$ | $0.898_{\pm0.002}$ | $0.902_{\pm0.005}$ | $0.913_{\pm0.005}$ | $0.916_{\pm0.001}$ | $0.934_{\pm0.001}$ |
| | 19 | $0.101_{\pm0.000}$ | $0.887_{\pm0.007}$ | $0.897_{\pm0.004}$ | $0.902_{\pm0.001}$ | $0.909_{\pm0.002}$ | $0.915_{\pm0.002}$ | $0.915_{\pm0.001}$ | $0.936_{\pm0.001}$ |
| | 20 | $0.100_{\pm0.000}$ | $0.903_{\pm0.005}$ | $0.904_{\pm0.003}$ | $0.908_{\pm0.002}$ | $0.910_{\pm0.004}$ | $0.920_{\pm0.002}$ | $0.921_{\pm0.001}$ | $0.935_{\pm0.002}$ |

| | | QualityMRI | | | | | | | |
|---|---|---|---|---|---|---|---|---|---|
| | $\alpha$ | 0.0 | 0.2 | 0.4 | 0.6 | 0.8 | 0.9 | 0.95 | 1.0 |
| Member $m$ | 1 | $0.624_{\pm0.033}$ | $0.602_{\pm0.020}$ | $0.602_{\pm0.020}$ | $0.602_{\pm0.020}$ | $0.602_{\pm0.020}$ | $0.602_{\pm0.020}$ | $0.624_{\pm0.033}$ | $0.532_{\pm0.105}$ |
| | 2 | $0.376_{\pm0.008}$ | $0.511_{\pm0.050}$ | $0.532_{\pm0.023}$ | $0.570_{\pm0.046}$ | $0.575_{\pm0.020}$ | $0.597_{\pm0.057}$ | $0.586_{\pm0.055}$ | $0.495_{\pm0.119}$ |
| | 3 | $0.624_{\pm0.008}$ | $0.608_{\pm0.015}$ | $0.672_{\pm0.027}$ | $0.629_{\pm0.026}$ | $0.618_{\pm0.015}$ | $0.624_{\pm0.055}$ | $0.656_{\pm0.042}$ | $0.570_{\pm0.075}$ |
| | 4 | $0.376_{\pm0.008}$ | $0.586_{\pm0.015}$ | $0.548_{\pm0.103}$ | $0.543_{\pm0.020}$ | $0.581_{\pm0.013}$ | $0.618_{\pm0.027}$ | $0.618_{\pm0.053}$ | $0.511_{\pm0.112}$ |
| | 5 | $0.624_{\pm0.008}$ | $0.608_{\pm0.027}$ | $0.645_{\pm0.035}$ | $0.602_{\pm0.042}$ | $0.597_{\pm0.070}$ | $0.629_{\pm0.013}$ | $0.651_{\pm0.038}$ | $0.570_{\pm0.101}$ |
| | 6 | $0.376_{\pm0.008}$ | $0.543_{\pm0.020}$ | $0.586_{\pm0.040}$ | $0.581_{\pm0.023}$ | $0.591_{\pm0.020}$ | $0.570_{\pm0.008}$ | $0.597_{\pm0.035}$ | $0.532_{\pm0.103}$ |
| | 7 | $0.624_{\pm0.008}$ | $0.618_{\pm0.042}$ | $0.624_{\pm0.020}$ | $0.608_{\pm0.020}$ | $0.618_{\pm0.030}$ | $0.640_{\pm0.042}$ | $0.629_{\pm0.013}$ | $0.511_{\pm0.124}$ |
| | 8 | $0.376_{\pm0.008}$ | $0.597_{\pm0.099}$ | $0.586_{\pm0.027}$ | $0.575_{\pm0.046}$ | $0.597_{\pm0.060}$ | $0.591_{\pm0.027}$ | $0.602_{\pm0.020}$ | $0.500_{\pm0.115}$ |
| | 9 | $0.624_{\pm0.008}$ | $0.640_{\pm0.020}$ | $0.651_{\pm0.055}$ | $0.683_{\pm0.046}$ | $0.613_{\pm0.023}$ | $0.645_{\pm0.013}$ | $0.629_{\pm0.013}$ | $0.522_{\pm0.107}$ |
| | 10 | $0.376_{\pm0.008}$ | $0.565_{\pm0.035}$ | $0.602_{\pm0.008}$ | $0.548_{\pm0.035}$ | $0.602_{\pm0.059}$ | $0.591_{\pm0.038}$ | $0.570_{\pm0.008}$ | $0.505_{\pm0.095}$ |
| | 11 | $0.624_{\pm0.008}$ | $0.629_{\pm0.026}$ | $0.624_{\pm0.020}$ | $0.608_{\pm0.027}$ | $0.591_{\pm0.033}$ | $0.613_{\pm0.066}$ | $0.581_{\pm0.013}$ | $0.565_{\pm0.068}$ |
| | 12 | $0.376_{\pm0.008}$ | $0.538_{\pm0.015}$ | $0.570_{\pm0.055}$ | $0.640_{\pm0.008}$ | $0.591_{\pm0.008}$ | $0.602_{\pm0.038}$ | $0.570_{\pm0.030}$ | $0.538_{\pm0.107}$ |
| | 13 | $0.624_{\pm0.008}$ | $0.688_{\pm0.053}$ | $0.613_{\pm0.013}$ | $0.677_{\pm0.023}$ | $0.656_{\pm0.008}$ | $0.602_{\pm0.042}$ | $0.618_{\pm0.042}$ | $0.559_{\pm0.100}$ |
| | 14 | $0.376_{\pm0.008}$ | $0.532_{\pm0.035}$ | $0.581_{\pm0.035}$ | $0.570_{\pm0.046}$ | $0.586_{\pm0.020}$ | $0.591_{\pm0.020}$ | $0.554_{\pm0.033}$ | $0.538_{\pm0.127}$ |
| | 15 | $0.624_{\pm0.008}$ | $0.634_{\pm0.040}$ | $0.651_{\pm0.008}$ | $0.586_{\pm0.050}$ | $0.651_{\pm0.059}$ | $0.629_{\pm0.023}$ | $0.618_{\pm0.050}$ | $0.559_{\pm0.112}$ |
| | 16 | $0.376_{\pm0.008}$ | $0.565_{\pm0.023}$ | $0.618_{\pm0.040}$ | $0.586_{\pm0.062}$ | $0.570_{\pm0.030}$ | $0.597_{\pm0.026}$ | $0.554_{\pm0.008}$ | $0.516_{\pm0.092}$ |
| | 17 | $0.624_{\pm0.008}$ | $0.624_{\pm0.040}$ | $0.677_{\pm0.035}$ | $0.629_{\pm0.047}$ | $0.618_{\pm0.027}$ | $0.591_{\pm0.053}$ | $0.624_{\pm0.040}$ | $0.554_{\pm0.066}$ |
| | 18 | $0.376_{\pm0.008}$ | $0.575_{\pm0.055}$ | $0.586_{\pm0.008}$ | $0.597_{\pm0.013}$ | $0.591_{\pm0.020}$ | $0.570_{\pm0.008}$ | $0.602_{\pm0.027}$ | $0.522_{\pm0.119}$ |
| | 19 | $0.624_{\pm0.008}$ | $0.608_{\pm0.050}$ | $0.656_{\pm0.046}$ | $0.634_{\pm0.055}$ | $0.629_{\pm0.040}$ | $0.624_{\pm0.038}$ | $0.618_{\pm0.040}$ | $0.527_{\pm0.077}$ |
| | 20 | $0.376_{\pm0.008}$ | $0.608_{\pm0.027}$ | $0.570_{\pm0.065}$ | $0.608_{\pm0.020}$ | $0.597_{\pm0.026}$ | $0.608_{\pm0.020}$ | $0.554_{\pm0.020}$ | $0.548_{\pm0.115}$ |

**Table 8: Accuracy per ensemble member of baselines** on ChaosNLI, CIFAR-10, and QualityMRI.

| | | **ChaosNLI** | | | |
|---|---|---|---|---|---|
| | | CreWra | CreEns$_\alpha$ | CreBNN | CreNet |
| | 1 | $0.658_{\pm0.013}$ | $0.671_{\pm0.011}$ | $0.680_{\pm0.013}$ | $0.525_{\pm0.051}$ |
| | 2 | $0.648_{\pm0.012}$ | $0.676_{\pm0.013}$ | $0.674_{\pm0.031}$ | $0.488_{\pm0.098}$ |
| | 3 | $0.641_{\pm0.020}$ | $0.676_{\pm0.013}$ | $0.517_{\pm0.079}$ | $0.542_{\pm0.030}$ |
| | 4 | $0.636_{\pm0.012}$ | $0.666_{\pm0.008}$ | $0.683_{\pm0.030}$ | $0.485_{\pm0.037}$ |
| | 5 | $0.651_{\pm0.032}$ | $0.667_{\pm0.011}$ | $0.533_{\pm0.112}$ | $0.554_{\pm0.051}$ |
| | 6 | $0.660_{\pm0.016}$ | $0.674_{\pm0.021}$ | $0.667_{\pm0.008}$ | $0.470_{\pm0.115}$ |
| | 7 | $0.637_{\pm0.014}$ | $0.677_{\pm0.009}$ | $0.675_{\pm0.012}$ | $0.518_{\pm0.036}$ |
| | 8 | $0.667_{\pm0.016}$ | $0.670_{\pm0.010}$ | $0.675_{\pm0.029}$ | $0.446_{\pm0.030}$ |
| **Member $m$** | 9 | $0.584_{\pm0.086}$ | $0.663_{\pm0.005}$ | $0.610_{\pm0.104}$ | $0.462_{\pm0.113}$ |
| | 10 | $0.663_{\pm0.035}$ | $0.674_{\pm0.029}$ | $0.668_{\pm0.017}$ | $0.531_{\pm0.064}$ |
| | 11 | $0.648_{\pm0.017}$ | $0.653_{\pm0.015}$ | $0.529_{\pm0.096}$ | $0.484_{\pm0.047}$ |
| | 12 | $0.660_{\pm0.016}$ | $0.662_{\pm0.016}$ | $0.684_{\pm0.019}$ | $0.549_{\pm0.071}$ |
| | 13 | $0.664_{\pm0.014}$ | $0.655_{\pm0.017}$ | $0.526_{\pm0.105}$ | $0.487_{\pm0.040}$ |
| | 14 | $0.650_{\pm0.011}$ | $0.654_{\pm0.006}$ | $0.668_{\pm0.030}$ | $0.505_{\pm0.039}$ |
| | 15 | $0.635_{\pm0.038}$ | $0.651_{\pm0.018}$ | $0.673_{\pm0.009}$ | $0.458_{\pm0.143}$ |
| | 16 | $0.652_{\pm0.026}$ | $0.654_{\pm0.007}$ | $0.616_{\pm0.108}$ | $0.486_{\pm0.012}$ |
| | 17 | $0.628_{\pm0.019}$ | $0.633_{\pm0.015}$ | $0.675_{\pm0.030}$ | $0.518_{\pm0.057}$ |
| | 18 | $0.652_{\pm0.009}$ | $0.605_{\pm0.021}$ | $0.598_{\pm0.109}$ | $0.519_{\pm0.031}$ |
| | 19 | $0.660_{\pm0.012}$ | $0.581_{\pm0.037}$ | $0.628_{\pm0.116}$ | $0.487_{\pm0.050}$ |
| | 20 | $0.651_{\pm0.013}$ | $0.480_{\pm0.074}$ | $0.536_{\pm0.116}$ | $0.560_{\pm0.043}$ |
| | | **CIFAR-10** | | | |
| | | CreWra | CreEns$_\alpha$ | CreBNN | CreNet |
| | 1 | $0.942_{\pm0.001}$ | $0.955_{\pm0.000}$ | $0.875_{\pm0.004}$ | $0.941_{\pm0.000}$ |
| | 2 | $0.944_{\pm0.001}$ | $0.954_{\pm0.000}$ | $0.877_{\pm0.005}$ | $0.943_{\pm0.000}$ |
| | 3 | $0.944_{\pm0.001}$ | $0.952_{\pm0.001}$ | $0.867_{\pm0.015}$ | $0.944_{\pm0.001}$ |
| | 4 | $0.943_{\pm0.002}$ | $0.952_{\pm0.000}$ | $0.881_{\pm0.003}$ | $0.942_{\pm0.001}$ |
| | 5 | $0.943_{\pm0.001}$ | $0.952_{\pm0.001}$ | $0.872_{\pm0.004}$ | $0.942_{\pm0.001}$ |
| | 6 | $0.943_{\pm0.000}$ | $0.952_{\pm0.000}$ | $0.869_{\pm0.006}$ | $0.942_{\pm0.002}$ |
| | 7 | $0.941_{\pm0.001}$ | $0.952_{\pm0.001}$ | $0.877_{\pm0.001}$ | $0.943_{\pm0.001}$ |
| | 8 | $0.942_{\pm0.001}$ | $0.951_{\pm0.001}$ | $0.880_{\pm0.006}$ | $0.943_{\pm0.002}$ |
| **Member $m$** | 9 | $0.943_{\pm0.001}$ | $0.951_{\pm0.000}$ | $0.879_{\pm0.005}$ | $0.944_{\pm0.001}$ |
| | 10 | $0.943_{\pm0.000}$ | $0.953_{\pm0.000}$ | $0.873_{\pm0.007}$ | $0.943_{\pm0.003}$ |
| | 11 | $0.942_{\pm0.001}$ | $0.953_{\pm0.001}$ | $0.882_{\pm0.004}$ | $0.943_{\pm0.000}$ |
| | 12 | $0.943_{\pm0.000}$ | $0.953_{\pm0.000}$ | $0.879_{\pm0.003}$ | $0.942_{\pm0.000}$ |
| | 13 | $0.943_{\pm0.000}$ | $0.953_{\pm0.001}$ | $0.874_{\pm0.000}$ | $0.943_{\pm0.002}$ |
| | 14 | $0.945_{\pm0.000}$ | $0.952_{\pm0.001}$ | $0.872_{\pm0.006}$ | $0.941_{\pm0.001}$ |
| | 15 | $0.942_{\pm0.002}$ | $0.948_{\pm0.000}$ | $0.856_{\pm0.019}$ | $0.942_{\pm0.001}$ |
| | 16 | $0.942_{\pm0.000}$ | $0.941_{\pm0.000}$ | $0.870_{\pm0.002}$ | $0.942_{\pm0.001}$ |
| | 17 | $0.942_{\pm0.001}$ | $0.930_{\pm0.001}$ | $0.854_{\pm0.035}$ | $0.942_{\pm0.002}$ |
| | 18 | $0.944_{\pm0.001}$ | $0.921_{\pm0.001}$ | $0.872_{\pm0.007}$ | $0.943_{\pm0.002}$ |
| | 19 | $0.943_{\pm0.001}$ | $0.906_{\pm0.002}$ | $0.873_{\pm0.001}$ | $0.942_{\pm0.001}$ |
| | 20 | $0.942_{\pm0.001}$ | $0.872_{\pm0.002}$ | $0.860_{\pm0.012}$ | $0.942_{\pm0.001}$ |
| | | **QualityMRI** | | | |
| | | CreWra | CreEns$_\alpha$ | CreBNN | CreNet |
| | 1 | $0.457_{\pm0.099}$ | $0.430_{\pm0.170}$ | $0.548_{\pm0.115}$ | $0.581_{\pm0.137}$ |
| | 2 | $0.452_{\pm0.139}$ | $0.409_{\pm0.163}$ | $0.484_{\pm0.126}$ | $0.559_{\pm0.112}$ |
| | 3 | $0.425_{\pm0.124}$ | $0.419_{\pm0.160}$ | $0.468_{\pm0.126}$ | $0.570_{\pm0.110}$ |
| | 4 | $0.419_{\pm0.126}$ | $0.446_{\pm0.187}$ | $0.548_{\pm0.115}$ | $0.575_{\pm0.102}$ |
| | 5 | $0.398_{\pm0.135}$ | $0.387_{\pm0.139}$ | $0.468_{\pm0.126}$ | $0.548_{\pm0.126}$ |
| | 6 | $0.403_{\pm0.117}$ | $0.398_{\pm0.158}$ | $0.554_{\pm0.118}$ | $0.554_{\pm0.084}$ |
| | 7 | $0.409_{\pm0.175}$ | $0.414_{\pm0.161}$ | $0.446_{\pm0.107}$ | $0.554_{\pm0.095}$ |
| | 8 | $0.430_{\pm0.119}$ | $0.441_{\pm0.182}$ | $0.554_{\pm0.118}$ | $0.559_{\pm0.133}$ |
| **Member $m$** | 9 | $0.392_{\pm0.137}$ | $0.419_{\pm0.181}$ | $0.473_{\pm0.110}$ | $0.586_{\pm0.141}$ |
| | 10 | $0.446_{\pm0.177}$ | $0.430_{\pm0.140}$ | $0.462_{\pm0.130}$ | $0.570_{\pm0.118}$ |
| | 11 | $0.425_{\pm0.137}$ | $0.414_{\pm0.153}$ | $0.554_{\pm0.118}$ | $0.602_{\pm0.107}$ |
| | 12 | $0.398_{\pm0.122}$ | $0.430_{\pm0.153}$ | $0.554_{\pm0.118}$ | $0.565_{\pm0.103}$ |
| | 13 | $0.398_{\pm0.130}$ | $0.430_{\pm0.140}$ | $0.446_{\pm0.107}$ | $0.575_{\pm0.090}$ |
| | 14 | $0.409_{\pm0.066}$ | $0.430_{\pm0.142}$ | $0.457_{\pm0.122}$ | $0.581_{\pm0.070}$ |
| | 15 | $0.409_{\pm0.146}$ | $0.414_{\pm0.132}$ | $0.468_{\pm0.126}$ | $0.624_{\pm0.133}$ |
| | 16 | $0.414_{\pm0.107}$ | $0.425_{\pm0.167}$ | $0.554_{\pm0.118}$ | $0.608_{\pm0.100}$ |
| | 17 | $0.419_{\pm0.139}$ | $0.414_{\pm0.119}$ | $0.554_{\pm0.118}$ | $0.581_{\pm0.091}$ |
| | 18 | $0.403_{\pm0.168}$ | $0.414_{\pm0.129}$ | $0.468_{\pm0.126}$ | $0.581_{\pm0.117}$ |
| | 19 | $0.425_{\pm0.151}$ | $0.414_{\pm0.073}$ | $0.554_{\pm0.118}$ | $0.565_{\pm0.142}$ |
| | 20 | $0.430_{\pm0.154}$ | $0.446_{\pm0.068}$ | $0.548_{\pm0.117}$ | $0.581_{\pm0.103}$ |

