# OpenReview forum: "Credal Prediction based on Relative Likelihood"
_NeurIPS.cc/2025/Conference — NeurIPS 2025 spotlight_

### Official Review · Reviewer_BkoJ · 2025-06-23

**Clarity:** 4
**Significance:** 3
**Originality:** 3
**Rating:** 5
**Confidence:** 5

**Summary:**

The paper proposes a method to learn credal set-valued predictors for classification problems. To train a credal set-valued predictors, standard approaches often use ensemble models, but those methods lack the notion of "learning" a credal set in the sense that there is no selection criteria going on. This is what makes this work shines.

The authors also proposed a smart initialisation approach to promote exploration during the training called ToBias.

**Questions:**

1. (about choosing alpha) Is there a way to propagate my epistemic uncertainty about the threshould to your framework? or does it even make sense to use some withheld dataset to choose some Pareto-ly undominated alphas? What is your opinion on this?

**Ethical Concerns:**

["NO or VERY MINOR ethics concerns only"]

**Final Justification:**

I remain very positive about this paper.

**Limitations:**

Limitations are clearly discussed.

**Paper Formatting Concerns:**

No.

**Quality:**

3

**Strengths And Weaknesses:**

Strength:
- the paper is well written. Concepts are easy to understand. The methods are principled.
- I particularly appreciate the authors not using predictive accuracy of any sort (collapsing your credal set e.g.) as a metric, instead use coverage and diversity/efficiency as the comparison metric.

Weakness:
- As with most credal set-valued or IP incorporated predictors, there aren't a lot of discussion to regression problems. It is not a big deal but seems like your method should also work for regression problem too? (not the ToBias I suppose but the relative likelihood business)

---

> ### Author Rebuttal · Authors · 2025-07-31
>
> We gratefully thank you for carefully reading our manuscript and providing a valuable review. We’re glad to hear that you appreciated our choice to evaluate using coverage and efficiency rather than collapsing the credal set to a single prediction. We believe these metrics better measure and reflect the goals of imprecise prediction in representing uncertainty.
>
> > **W1:** As with most credal set-valued or IP incorporated predictors, there aren't a lot of discussion to regression problems. It is not a big deal but seems like your method should also work for regression problem too? (not the ToBias I suppose but the relative likelihood business)
>
> We agree that the core idea, particularly the relative likelihood-based construction, should, in principle, extend to regression settings as well. We are currently exploring this direction in follow-up work. That said, applying our framework to regression requires specifying the form of the predictive distributions (e.g., Gaussian, etc.) and how to define credal sets over them. Additionally, metrics such as coverage and efficiency are less straightforward in regression and would need to be adapted or newly defined to reflect meaningful uncertainty representation.
>
> We appreciate the suggestion and inspiration and we will report on it in follow-up research.
>
> > **Q1:** (about choosing alpha) Is there a way to propagate my epistemic uncertainty about the threshould to your framework? or does it even make sense to use some withheld dataset to choose some Pareto-ly undominated alphas? What is your opinion on this?
>
> We are not entirely sure what you mean by (epistemic) uncertainty about $\alpha$, because there is no “true” $\alpha$ that we seek to estimate. Instead, $\alpha$ is used as a control parameter to find a good compromise between coverage and efficiency (very much like a p-value in statistics, which controls the compromise between type-1 and type-2 error in hypothesis testing). What is indeed uncertain is the (expected) coverage and efficiency for a given $\alpha$, and using extra data for estimating these quantities certainly makes sense (we are currently computing them on test data).
>
> If we compute the entire Pareto-curve, as we do in the paper, then the user can choose the most appropriate point on this curve (and hence implicitly the corresponding $\alpha$). Thus, there is no need to specify an $\alpha$ beforehand.
>
> However, computing the entire Pareto-curve might be considered too costly. Then, the best point on the curve should be found in a more targeted manner. One could imagine, for example, that this is done in an interactive process with the human in the loop. The system could make a suggestion (e.g., a specific point on the curve for an initial $\alpha$), the user may provide feedback, and the system can adapt the $\alpha$ based on that feedback. This looks like a very interesting direction for future work, thanks for raising this topic.

---

> > ### Comment · Reviewer_BkoJ · 2025-08-08
> >
> > Thanks for your clarification and I remain positive about this paper.

---

### Official Review · Reviewer_gS33 · 2025-06-28

**Clarity:** 4
**Significance:** 4
**Originality:** 3
**Rating:** 5
**Confidence:** 3

**Summary:**

The paper proposes a method for generating credal predictors within a machine learning framework, with a focus on neural networks. The approach selects plausible models based on relative likelihood. To promote diversity among predictors, the authors introduce a novel initialisation method called ToBias. This strategy ensures that the ensemble begins in a state of full uncertainty, represented by initial predictions located at the vertices of the probability simplex. As training proceeds, the ensemble’s knowledge increases, leading to a progressive shrinking of the credal set.

**Questions:**

None

**Ethical Concerns:**

["NO or VERY MINOR ethics concerns only"]

**Final Justification:**

I was already inclined to accept this paper, and the authors’ responses to the other reviewers’ critiques seem to have had a positive effect.

**Limitations:**

Yes

**Quality:**

3

**Strengths And Weaknesses:**

The paper presents a structured approach to generating credal predictors using neural networks, with model selection based on relative likelihood. The method is technically sound and supported by experimental results. A key component is the ToBias initialisation strategy, which assigns initial predictions at the vertices of the probability simplex to promote diversity. While this is not a major conceptual innovation, it is a well-considered design choice that supports the overall framework. The authors provide a balanced evaluation, carefully stating the scope and limitations of their contribution.

The writing is clear and the organisation is logical. The paper presents its arguments methodically, with sufficient detail to support reproducibility. The clarity of the exposition makes the technical material accessible to readers familiar with the field.

The work is relevant to ongoing research on uncertainty quantification in neural networks. The proposed method is practical and may be of interest to those working with ensemble learning or imprecise probabilities. It addresses a persistent challenge in the area and offers a structured solution that can inform future research.

Although the paper does not introduce an entirely new method, it coherently combines known techniques and applies them to a problem of practical importance. In line with the NeurIPS 2025 Reviewer Guidelines, the work provides a meaningful contribution through thoughtful adaptation and analysis of existing methods.

---

> ### Author Rebuttal · Authors · 2025-07-31
>
> We thank you for the thoughtful and encouraging feedback. We appreciate the recognition of our structured approach to credal prediction, the practicality of the ToBias initialization strategy, and the clarity of our exposition.
> We’re especially glad that you found our evaluation balanced and reproducible, and that our approach was seen as a meaningful contribution in line with the NeurIPS guidelines. Our aim was indeed to address a persistent challenge in uncertainty quantification with a principled and implementable solution.

---

### Official Review · Reviewer_F5Am · 2025-07-03

**Clarity:** 2
**Significance:** 3
**Originality:** 3
**Rating:** 5
**Confidence:** 2

**Summary:**

This work deals with the task of learning neural network ensembles capable of cautious decision-making. The authors propose CreRL, which constructs this ensemble by learning models satisfying the relative likelihood constraint: each model’s (conditional) likelihood cannot be less than a given percent (\alpha) of that of the maximum likelihood solution. To ensure diversity the authors (1) use early stopping, that is, stopping training once the neural network satisfies the relative likelihood constraint; (2) they propose an initialization strategy call ToBias, which sets the one of final layer biases to a high value, e.g., 100; models with this an initialization cover the probability simplex better, approximating prior ignorance. Inference is performed by computing the predictive probabilities under all the models and taking the minimum and maximum values to construct a closed convex credal set. These credal sets are evaluated on their efficiency and their coverage; the former measuring the smallness of the credal set, while the latter measuring whether the ground truth probability lies in the credal set. The authors perform this evaluation on the ChaosNLI, CIFAR-10, and QualityMRI datasets. The proposed method CreRL achieves efficiency and coverage. Moreover, varying the hyperparameter \alpha allows it to tradeoff one for the other. The authors also evaluate out of distribution detection, using CIFAR-10 as the in-distribution data set, and SVHN, Places, CIFAR-100, FMNIST, and ImageNet as out of distribution data sets. They find that CreRL with \alpha = 1 performs on par or better than the best performing baselines.

**Questions:**

The relative likelihood credal sets seem to be closely related to Rashomon sets in Rudin et al. (2024). Does CreRL approximate the Rashomon set of neural predictors?

Rudin, Cynthia, et al. "Position: Amazing Things Come From Having Many Good Models." ICML 2024

**Ethical Concerns:**

["NO or VERY MINOR ethics concerns only"]

**Final Justification:**

The authors have addressed all my concerns. So, I have updated my score to 5 (accept)

**Limitations:**

yes

**Quality:**

3

**Strengths And Weaknesses:**

Strengths
1. Cautious decision making is important in high-stakes domains like healthcare. This method allows highly expressive neural network models to be used with a principled, cautious decision-making strategy.
2. The proposed method CreRL is simple and easy to implement
3. Evaluation of credal set-based methods using coverage and efficiency metrics from conformal prediction literature.

Weaknesses
1. Section 5 Empirical results says that: “Our method (CreRL) with \alpha 1 performs either the best or is on par with the best methods in OoD detection, except for the SVHN dataset, where the CreWra approach has a small advantage.” But, it is unclear why CreWra has an advantage in the SVHN data set.
2. In the OOD evaluation, the best performing model has \alpha = 1. This seems to be the same as an ensemble of fully trained maximum likelihood models, indicating that credal sets based on the relative likelihood constraint are not useful in OoD detection

---

> ### Author Rebuttal · Authors · 2025-07-31
>
> Thank you very much for your careful reading of our manuscript and for your valuable feedback. We appreciate your recognition of the relevance and usefulness of our approach for decision-making.
>
> > **W1, W2:** Section 5 Empirical results says that: “Our method (CreRL) with \alpha 1 performs either the best or is on par with the best methods in OoD detection, except for the SVHN dataset, where the CreWra approach has a small advantage.” But, it is unclear why CreWra has an advantage in the SVHN data set.
> In the OOD evaluation, the best performing model has \alpha = 1. This seems to be the same as an ensemble of fully trained maximum likelihood models, indicating that credal sets based on the relative likelihood constraint are not useful in OoD detection
>
> Indeed, our method performs better for some datasets, e.g. FMNIST, and worse for SVHN, as you rightfully pointed out. Please note, however, that the difference on SVHN is rather small (0.009) and not statistically significant (in light of the standard deviation). This is consistent with our claim that our method performs comparably on the OoD task to the evaluated baselines. To analyze the performance on the SVHN dataset further, we present the mean epistemic uncertainty (EU) values on the in-distribution (iD) data (CIFAR-10) and out-of-distribution (OoD) data (SVHN) in the table below.
>
> | $\alpha$ |   0.2  |  0.4  | 0.6   |  0.8  |  0.9  | 0.95 | 1.0  | CreWra |
> |---------|--------|-------|--------|--------|-------|--------|-------|------------|
> | iD       | 2.29 | 1.74 | 1.34 | 0.95 | 0.79 | 0.70 | 0.45 |    0.38   |
> | OoD   | 2.60 | 2.49 | 2.53 | 2.42 | 2.34 | 2.25 | 2.23 |    2.22   |
>
> Notably, we observe that increasing $\alpha$ shrinks the credal sets for iD data more than it does for OoD data, hence the gap is highest for $\alpha = 1$, as evidenced by the mean EU values. This results in an increased ability to discriminate between iD and OoD data. The pattern above can also be observed for other data sets. Broadly speaking, reducing $\alpha$ (purposely) increases the uncertainty for iD data and makes it less distinguishable from OoD data (note that $\alpha = 0$ would produce the same vacuous credal sets for both iD and OoD data). This explains why $\alpha = 1$ is best for discriminating between iD and OoD data — which, of course, does not mean that it also yields the best or most faithful representation of uncertainty. Note that with $\alpha=1$, only models with a high (relative) likelihood are included, similar to the “regular” ensemble learning used in the credal wrapper method, which may explain their similar performance in general.
>
> As for the claim that credal sets based on the relative likelihood are not useful in OoD detection, we kindly disagree. A major benefit of our approach is that it offers the user the flexibility to choose a particular relative likelihood threshold. Thus, the user could opt for a large $\alpha$, thereby increasing efficiency and OoD performance, or a smaller $\alpha$, increasing coverage, but sacrificing efficiency and OoD performance. While our aim is not to obtain OoD performance, our method still performs competitively with smaller $\alpha$ values such as 0.8.
>
> > **Q1:** The relative likelihood credal sets seem to be closely related to Rashomon sets in Rudin et al. (2024). Does CreRL approximate the Rashomon set of neural predictors?
>
> This is an interesting question! Indeed, there seems to be a strong connection to Rashomon sets. Both Rashomon sets and relative likelihood credal sets describe the set of all models with a performance above some threshold [1]. The purpose is quite different though. While we are interested in uncertainty quantification and diversity of predictions, Rashomon sets focus on interpretability and (syntactic) model diversity (because interpretability is compromised if there are many different models that describe the data equally well).
>
> Despite the different objectives and motivations, they both have to address the same problem: the approximation of the set of plausible models [2]. In the approach proposed by us, ToBias initialization is used as a way to promote diverse outputs, whereas in the Rashomon set literature other techniques are used. An interesting avenue for future research could explore whether such techniques would also be beneficial for credal set generation. Thank you for bringing these works to our attention. We will make sure to highlight this connection in our paper.
>
> [1] On the Existence of Simpler Machine Learning Models. Lesia Semenova, Cynthia Rudin, and Ronald Parr. 2022.
>
> [2] Rashomon Sets for Prototypical-Part Networks: Editing Interpretable Models in
> Real-Time. Donnelly et al. 2025.

---

### Official Review · Reviewer_vaSE · 2025-07-03

**Clarity:** 3
**Significance:** 2
**Originality:** 2
**Rating:** 5
**Confidence:** 4

**Summary:**

This paper presents a method to obtain a credal set prediction and introduces $\alpha$, a threshold on relative likelihood to balance between coverage and efficiency. The credal set prediction is based on ensemble of model predictions. The approach is empirically evaluated on three datasets for AUC curve on efficiency vs. coverage while also including a study of OOD detection

**Questions:**

Why is there a need to do thresholding on relative likelihood? Let’s say, we train 10 models on CIFAR-10 and choose $\alpha$ to be 0.9, how many models will be dropped?

**Ethical Concerns:**

["NO or VERY MINOR ethics concerns only"]

**Final Justification:**

I appreciate the author's response. It has cleared my concerns and confusions. Therefore, I raise my score.

**Limitations:**

yes

**Quality:**

2

**Strengths And Weaknesses:**

Strengths:
-	The method is relatively simple and presented very clearly.
-	The use of ToBias approach to get a wide credal set at the start of training is commendable.

Weaknesses:
-	The experimental analysis lacks in depth. More datasets could be explored.
-	It is claimed that the proposed outperforms existing methods but it is not evident in the results. The proposed approach clearly lacks in coverage (and by extension accuracy). The OOD results are comparable to the existing methods and do not outperform.
-	A clear study of the accuracy metric is missing.
-	The only novel thing, as per my understanding, is the use of thresholding on relative likelihood to select the best models. While interesting, it does not make a case for NeurIPS level publication.

---

> ### Author Rebuttal · Authors · 2025-07-31
>
> We gratefully thank you for reading our manuscript and providing a valuable review.
>
> > **W1:** The experimental analysis lacks in depth.
>
> We would like to clarify that our experimental analysis focuses on evaluating the quality of the uncertainty representation through the trade-off between coverage and efficiency. The objective is to construct credal sets that contain the ground-truth distribution (coverage) while being as informative (small) as possible (efficiency). Ideally, both metrics would be close to 1, corresponding to the top-right corner in the Pareto plot (Figure 4). However, this ideal is rarely reachable in practice, as improving efficiency often comes at the cost of reduced coverage, and vice versa.
>
> To illustrate this trade-off, we provide experiments on three diverse datasets and compare our method to four baselines. As also positively noted by Reviewers gS33 and BkoJ, we believe this provides a well-balanced and comprehensive evaluation of our approach, and in fact goes beyond the experimental scope of comparable credal prediction papers. Additionally, we include an out-of-distribution (OoD) detection task as a sanity check to demonstrate that our method captures meaningful uncertainty, even though improving downstream task performance is not the primary aim of our work.
>
> We hope this clarifies our experimental design and would appreciate your feedback on which aspects are still lacking.
>
> > **W2:** More datasets could be explored.
>
> As noted in the paper, our evaluation relies on datasets with first-order annotations in the form (ground-truth) probability distributions over classes, which are essential for computing the coverage metric. Such datasets are difficult to find, but we identified and used three that support meaningful evaluation, demonstrating the effectiveness and adaptability of our approach. We gladly welcome suggestions for additional datasets to expand our empirical analysis in future work.
>
> > **W3:** It is claimed that the proposed outperforms existing methods but it is not evident in the results. The proposed approach clearly lacks in coverage (and by extension accuracy). - A clear study of the accuracy metric is missing.
>
> Coverage: Based on the results in Figure 4 (and Table 6 in the appendix), we do not understand the concern regarding lack of coverage in our approach. Quite the opposite, one of the key strengths of our method is the ability to control the trade-off between coverage and efficiency by adjusting the threshold $\alpha$. This allows us to move along the Pareto front, outperforming all baseline methods across a range of settings. In fact, ours is the only method that reaches high coverage regions by varying $\alpha$, while others remain stuck in low coverage, often failing to include the ground truth.
>
> Accuracy: We are not entirely sure what you mean by accuracy. Typically, accuracy refers to the fraction of correct predictions in a setting where both prediction and ground truth are class labels associated with an instance x. In our setting, however, the ground truth is a probability distribution (associated with x), and the prediction a set of such distributions (a credal set). In this setting, it is not at all obvious how accuracy should be defined.
>
> That said, please note that we do report standard accuracy scores in Tables 7 and 8 in the appendix. We did this as a kind of sanity check, to make sure that the individual models used in our ensembles are well trained. As these are still probabilistic models, we considered a prediction correct if the class with the highest predicted probability coincides with the class with the highest true probability.
>
> When considering these tables, note that the individual ensemble members for lower $\alpha$ values may also have lower accuracy values. This is expected, and even desirable, as we wish to include models with a lower (relative) likelihood, which will in turn have an accuracy that is lower than the MLE (the first row of the tables). This is in line with our method design.
>
> > **W4:** The OOD results are comparable to the existing methods and do not outperform.
>
> Indeed, the OoD results are comparable to the baselines as also claimed in the paper. We would like to emphasize again that our goal is not to show superior performance on this downstream task, but to demonstrate that a competitive OoD detection is possible with our method. In particular, a major benefit of our approach is that it offers the user the flexibility to choose a particular relative likelihood threshold. Thus, the user could opt for a large $\alpha$, thereby increasing efficiency and OoD performance, or a smaller $\alpha$, thereby increasing coverage but sacrificing efficiency and OoD performance (see Figure 4 in the paper).
>
> To further highlight the flexibility of our approach and the comparable results to credal wrapper, we analyze the mean epistemic uncertainty (EU) values on the in-distribution (iD) data (CIFAR-10) and one OoD dataset (ImageNet) in the table below.
>
> | $\alpha$ |   0.2  |  0.4  | 0.6   |  0.8  |  0.9  | 0.95 | 1.0  | CreWra |
> |---------|:--------:|:-------:|:--------:|:--------:|:-------:|:--------:|:-------:|:------------:|
> | iD       | 2.29 | 1.74 | 1.34 | 0.95 | 0.79 | 0.70 | 0.45 |    0.38   |
> | OoD   | 2.78 | 2.60 | 2.47 | 2.30 | 2.20 | 2.14 | 1.96 |    1.83   |
>
> Notably, increasing $\alpha$ shrinks the credal sets for iD data more than it does for OoD data, hence the gap is highest for $\alpha = 1$, as evidenced by the mean EU values. This results in an increased ability to discriminate between iD and OoD data. Broadly speaking, reducing $\alpha$ (purposely) increases the uncertainty for iD data and makes it less distinguishable from OoD data (note that $\alpha = 0$ would produce the same vacuous credal sets for both iD and OoD data). Intuitively, increasing uncertainty and including predictions with lower likelihood indeed means less exploitation of the iD data. This explains why $\alpha = 1$ is best for discriminating between iD and OoD data -- which, of course, does not mean that it also yields the best or most faithful representation of uncertainty. Note that with $\alpha=1$, only models with a high (relative) likelihood are included, similar to the “regular” ensemble learning used in the credal wrapper method, which may explain their similar performance in general.
>
>
> > **Q1:** The only novel thing, as per my understanding, is the use of thresholding on relative likelihood to select the best models.
> > Why is there a need to do thresholding on relative likelihood? Let’s say, we train 10 models on CIFAR-10 and choose  to be 0.9, how many models will be dropped?
>
> This can’t be said in general. Particularly, our algorithm does not involve a post-hoc selection step, hence trained models will not be dropped. In fact, the threshold $\alpha$ (e.g. 0.9) is used to stop the training process once the model reaches the specified threshold. Consequently, all models have a relative likelihood greater or equal to 0.9, so that none of them is dropped (see Algorithm 1 in Section 4). Recall that the ensemble is conceived as a discrete approximation of the true $\alpha$-cut of the relative likelihood function. The more ensemble members we find inside that cut, the better the approximation will be.
>
> Let us reiterate the main purpose of our method, namely to approximate the set of all plausible models, and hence the set of all plausible (probability) predictions, where “plausible” is operationalized in terms of the $\alpha$-constraint on relative likelihood: a model is deemed plausible if its relative likelihood exceeds $\alpha$.
>
> In addition to this conceptual contribution, we provide a practical and scalable framework for training credal predictors in complex machine learning settings. The novel thing here is clearly not the thresholding, but rather the training procedure for constructing ensembles. It is designed in such a way that an accurate approximation of the $\alpha$-cut can be obtained from the ensemble members in an efficient and effective way (which, as already said, implies that as few models are discarded as possible). Standard ensemble learning is unsuitable for that purpose.
>
> We hope to have adequately addressed your questions and remarks. If you have any further questions, we would be happy to discuss them. Otherwise, we would appreciate it if you would reconsider your score.

---

### Note · Authors · 2025-08-14

We would like to thank all reviewers for taking the time to read and evaluate our work so carefully. Reviewers F5Am, gS33 and BkoJ recognised the usefulness, originality and scope of our contribution, and asked interesting questions about its connections to other fields, such as Rashomon sets and regression. These comments have inspired ideas for future research that build on the present work. F5Am also asked about out-of-distribution (OoD) performance, which gave us the opportunity to clarify the interpretation of the OoD results further.

Unlike the other reviewers, reviewer vaSE found the experimental evaluation unclear (e.g. they asked for a detailed accuracy evaluation). As most of their concerns do not directly relate to the purpose of our paper, we took the opportunity to clarify our objectives more explicitly and explain our evaluation methodology in greater detail. However, we are still unsure about the exact nature of some of these concerns. For instance, an evaluation based on the accuracy metric would not be applicable (or meaningful) when assessing our work on credal prediction. As the reviewer did not engage in further discussion with us, we believe that we have sufficiently addressed their remarks and answered their questions.

In summary, the reviewers agree that our work makes a significant and substantial contribution to credal prediction in machine learning. They acknowledge its novelty and theoretical soundness, as well as the improvements it brings to empirical results. We believe these advances will have a positive impact on the NeurIPS community.

---

### Decision · Program_Chairs · 2025-09-17

**Decision:**

Accept (spotlight)

**Comment:**

This paper proposes a principled framework for credal prediction based on relative likelihood, introducing novel methods to construct diverse ensembles that generate set-valued probability predictions with controllable trade-offs between coverage and efficiency. The reviewers converged on the view that the contribution is technically sound, novel in its use of relative likelihood for defining plausible model sets, and clearly written, with empirical validation across multiple datasets. While one reviewer initially raised concerns about the experimental evaluation, these were convincingly addressed in the rebuttal, and the reviewer increased their score accordingly. Other reviewers emphasized the clarity, structured design, and potential impact of the work in advancing uncertainty quantification, though they noted that connections to related areas (such as Rashomon sets) and extensions to regression could be explored further in future research. Overall, the rebuttal strengthened the consensus around the paper’s novelty, soundness, and relevance, and I recommend to accept this paper as a poster.